# Upper Cervical Manipulation and Manual Massage Do Not Modulate Sympatho-Vagal Balance or Blood Pressure in Women: A Randomized, Placebo-Controlled Clinical Trial

**DOI:** 10.3390/healthcare13202554

**Published:** 2025-10-10

**Authors:** Estêvão Rios Monteiro, Linda S. Pescatello, Gustavo Henrique Garcia, Alexandre Gonçalves de Meirelles, Francine de Oliveira, Rafael Cotta de Souza, Leandro Alberto Calazans Nogueira, Agnaldo José Lopes, Daniel Moreira-Gonçalves

**Affiliations:** 1Graduate Program in Rehabilitation Science, Centro Universitário Augusto Motta (PPGCR/UNISUAM), Rio de Janeiro 21032-060, Brazil; alexandremeirelles@souunisuam.com.br (A.G.d.M.); rafaesouza@souunisuam.com.br (R.C.d.S.); leandronogueira@souunisuam.com.br (L.A.C.N.); alopes@souunisuam.com.br (A.J.L.); 2Graduate Program in Biopsychosocial Health, Centro Universitário Augusto Motta (PPGBPS/UNISUAM), Rio de Janeiro 21032-060, Brazil; 3Department of Kinesiology, University of Connecticut, Storrs, CT 06269, USA; linda.pescatello@uconn.edu; 4Undergraduate Program in Physical Therapy, Centro Universitário Augusto Motta (UNISUAM), Rio de Janeiro 21041-020, Brazil; gustavogarcia@souunisuam.com.br; 5Postgraduate Program in Physical Education, Universidade Federal do Rio de Janeiro, Rio de Janeiro 21941-599, Brazil; francinerdeoliveiras@gmail.com; 6Physiotherapy Department, Federal Institute of Rio de Janeiro (IFRJ), Rio de Janeiro 21710-240, Brazil; 7Medical Sciences Postgraduation Program, School of Medical Sciences, Universidade do Estado do Rio de Janeiro (UERJ), Rio de Janeiro 20551-030, Brazil; 8Research Center in Physical Activity, Health and Leisure, Faculty of Sport, University of Porto, 4200-450 Porto, Portugal; danielmgon@fade.up.pt; 9Laboratory for Integrative and Translational Research in Population Health, 4200-450 Porto, Portugal

**Keywords:** spine manipulation, HVLA, myofascial release therapy, manual therapy, randomized controlled trial

## Abstract

**Objectives**: To compare the acute effects of upper cervical manipulation (CM) and manual massage (MM) to simulated CM (Sham) and Control conditions (Control) on heart rate variability (HRV) and blood pressure (BP) responses in women with non-elevated BP. **Methods**: A single-blind, four-arm, parallel-group, randomized, crossover, placebo-controlled trial recruited 15 apparently healthy women with non-elevated BP who visited the lab on four occasions with 48 h intervals to ensure adequate washout between interventions. A Latin square randomization approach was employed to assign participants to one of four experimental conditions: (1) Control: Rest without intervention; (2) CM: Bilateral high-velocity, low-amplitude manipulation of the upper cervical spine (C0–C2); (3) MM: A single 120 s session of MM release applied unilaterally to the anterior and posterior thigh, posterior lower leg, and lumbar musculature; or (4) Sham: Mimicking the positioning used in CM without the application of thrust manipulation. In each experiment, HRV, systolic and diastolic BP were measured at rest (Baseline) and every 15 min for 60 min after each intervention. All procedures were performed in the morning to avoid any confounding circadian rhythm effect on HRV and BP. **Results**: We found significant increases within conditions for RMSSDms (Control: Post-0 (*p* = 0.032), Post-15 (*p* = 0.023); Sham: Post-15 (*p* = 0.014); CM: Post-15 (*p* = 0.027)); SDNNms (Control: Post-45 (*p* = 0.037); CM: Post-45 (*p* = 0.014) and Post-60 (*p* = 0.019)); PNN50% (CM: Post-0 (*p* = 0.044), Post-15 (*p* = 0.044) and Post-45 (*p* = 0.019)); LF Power (ms2) (CM: Post-60 (*p* = 0.001)), and LF/HF ratio (MM: Post-60 (*p* = 0.022). **Conclusions**: Although no statistically significant between-condition differences were detected, within-condition changes with moderate-to-large effect sizes suggest potential clinical relevance of CM and MM. These preliminary findings emphasize the importance of effect sizes and may indicate greater translational significance in populations with non-elevated cardiovascular risk.

## 1. Introduction

Hypertension affects approximately 31.1% of the global adult population, highlighting the importance of exploring different strategies to regulate blood pressure (BP) as a key public health priority [1]. Data from the Non-Communicable Diseases (NCD) Risk Factor Collaboration [2] show that the prevalence of hypertension among individuals 30–79 years has doubled between 1990 and 2019, increasing to 331 million women. Hypertension is a leading risk factor for cardiovascular disease [3], and it remains a critical global health challenge [4]. Acute reductions in systolic BP ranging from 5 to 8 mmHg have been associated with clinically meaningful cardiovascular risk attenuation, representing a relevant non-pharmacological strategy for early hemodynamic modulation [5]. Thus, strategies to acutely reduce the BP have been explored. For instance, Monteiro et al. [6] reported a reduction in systolic BP of 13 ± 7 mmHg among normotensive individuals following an upper cervical spinal manipulation (CM).

Although no direct relationship has been established between the CM and the reduction in the BP, there appears to be a compensatory interplay between hemodynamic and autonomic responses following sympathetic-vagal stimulation. The autonomic nervous system (ANS), which includes the parasympathetic and sympathetic divisions, is an integral part of the nervous system’s role in maintaining homeostasis by regulating the functions of cells, tissues, and organs [7]. Control of this system is exerted by higher brain centers, including the limbic system, hypothalamus, and specific brainstem nuclei [7]. Cardiac ANS activity is commonly assessed indirectly using non-invasive markers such as heart rate variability (HRV) [8]. HRV is a non-invasive and sensitive marker of cardiac autonomic regulation, reflecting the dynamic interplay between sympathetic and parasympathetic influences on heart rhythm [8]. In this context, Lastova et al. [9], Picchiottino et al. [10], and Jupin et al. [11] demonstrated changes in sympatho-vagal balance after manual massage (MM) (e.g., myofascial release protocol) and joint manipulation, respectively.

Physical therapists, chiropractors, and osteopaths frequently utilize spinal manipulations to restore musculoskeletal function [12,13]. Nevertheless, tissue mobilization techniques (e.g., MM) are often used as an adjunct to treatment protocols. Previous literature suggests distinct neurophysiological effects in response to CM [14,15,16] and MM [17]. However, their effects are not yet fully understood, especially regarding BP and HRV responses. A limited studies examining the effects of upper CM on BP responses [18,19,20,21,22] and its influence on cardiac electrophysiology, HRV and electrocardiogram parameters [19,21]. The MM literature includes a few studies examining the impact of tissue mobilization on BP responses [23,24]. Notably, only one of the studies has focused exclusively on women participants [23], and the number of studies addressing the effects on HRV [25,26,27] remains markedly limited. Thus, this phenomenon highlights an important gap in the literature that the present study aims to address.

Accordingly, this study aimed to compare the acute effects of CM and MM to simulated CM (Sham) and Control on HRV and BP responses in asymptomatic individuals. The present study had two initial hypotheses. The first hypothesis suggests that the CM would demonstrate better sympatho-vagal balance compared to the Control and Sham. This hypothesis is based on the anatomical and physiological context of vagus nerve stimulation [28], specifically the proximal branch that passes bilaterally through the cervical region [29]. The second hypothesis proposes a higher magnitude of systolic BP reduction through the MM by myofascial release condition compared to the Control and Sham. This hypothesis is based on physiological mechanisms previously proposed for a similar technique performed with foam rolling, which showed higher concentrations of nitric oxide [30], improved arterial perfusion in the lateral thigh region [31], and promoted reductions in systolic BP [32,33].

## 2. Materials and Methods

### 2.1. Study Design

A single-blind, four-arm, parallel-group, randomized, crossover, placebo-controlled trial was conducted in accordance with the Consolidated Standards of Reporting Trials (CONSORT) [34] and Standard Protocol Items: Recommendations for Interventional Trials (SPIRIT) [35] guidelines. The protocol received approval from the Research Ethics Committee of the Augusto Motta University Center (IRB No. 7.058.455) and was prospectively registered in the Brazilian Registry of Clinical Trials (RBR-47274gx; WHO Universal Trial Number: U1111-1252-3077). All the participants signed the informed consent form before participating.

### 2.2. Participants

The trial was conducted at the Musculoskeletal Performance Laboratory at Graduate Program in Rehabilitation Science (PPGCR/UNISUAM), Rio de Janeiro, Brazil. Participants were recruited through a convenience sampling strategy, aiming to reduce heterogeneity and control for physiological differences that could otherwise act as confounding factors. Recruitment was conducted by invitation, following an announcement at the Augusto Motta University Center (undergraduate and graduate programs) and disseminated via social networks. Participants were women aged between 19 and 44 years, presenting resting systolic and diastolic BP values within <120/70 mmHg (Table 1), respectively, classified as non-elevated BP according to the 2024 European Society of Cardiology guidelines [36] and engaging in at least 300 min of physical activity per week. Participants were excluded if they (1) demonstrated any clinical or functional alteration of the basilar artery during the initial vertebrobasilar integrity screening and/or (2) reported habitual intake of caffeine-based supplements, or consumed any other substances known to influence basal metabolic rate or autonomic function. These criteria ensure participant safety and minimize potential confounding factors during the application of spinal manipulative procedures. Participation in the study was entirely voluntary. Individuals who decline participation or are unable to complete the study for any reason were classified as dropouts.

### 2.3. Procedures

Participants attended four laboratory sessions, each separated by a 48 h interval to minimize potential carryover effects. A Latin square randomization approach was employed to assign participants to one of four experimental conditions: (1) CM: bilateral high-velocity, low-amplitude manipulation of the upper cervical spine (C0–C2); (2) MM: a single 120 s session of MM by myofascial release applied unilaterally to the anterior and posterior thigh, posterior lower leg, and lumbar musculature; (3) Sham: mimicking the positioning used in CM without the application of thrust manipulation; or (4) Control: rest without intervention. Participants were informed that they would undergo a CM, a Sham, or an MM, and that all procedures were potentially therapeutic. Following this briefing, participants were directed to the licensed physical therapist responsible for administering the assigned intervention. Both the participants and the outcome assessors remained blinded to group allocation, whereas only the intervention administrator was not blinded, thereby maintaining a single-blind design throughout the study. Conversely, the investigator administering the interventions was aware of the specific condition assigned to each participant on each experimental day.

For each experimental session, HRV, systolic and diastolic BP were measured at rest (Baseline) and every 15 min for 60 min after each intervention (Post-0, Post-15, Post-30, Post-45, and Post-60). All procedures were performed in the morning to avoid any confounding effect of circadian rhythm on HRV and BP. Regarding temperature, ambient temperature was kept constant at levels unlikely to activate thermoregulatory responses, thereby minimizing potential interference with HRV measurements.

Participants were recommended to maintain their eating and sleeping habits during the ten days of data collection. Prior to the study, all participants received verbal and written explanations of all study procedures and completed the Physical Activity Readiness Questionnaire. Following the completion of the baseline assessment, the primary evaluator (Examiner 1) exited the testing area to maintain blinding regarding condition allocation. Subsequently, a licensed physical therapist (Examiner 2), trained and experienced, entered to administer the CM or the corresponding Sham or MM intervention, as determined by the randomization condition. Upon completion of the intervention, Examiner 2 left the room, allowing Examiner 1 to re-enter and conduct the identical post-intervention assessments using the same standardized protocols implemented at baseline.

### 2.4. Randomization

Randomization and allocation concealment were performed using sequentially numbered, opaque, sealed envelopes (1–15), each containing a pre-determined assignment card. An independent researcher, who had no involvement in the assessment or delivery of interventions, was responsible for implementing the allocation. Envelopes were opened only after participants completed informed consent and baseline testing. Each participant was assigned a unique study ID to ensure anonymity. To assess the effectiveness of blinding between the CM and Sham conditions, participants were asked to indicate which intervention they believed they had received following post-intervention HRV and BP assessments. Additionally, they were asked to report their perception of the intervention’s efficacy by answering the question: “Do you believe the intervention you received has a therapeutic effect?”. These assessments were conducted by a blinded research assistant uninvolved in any other stage of the trial.

### 2.5. Interventions

Cervical Manipulation (CM): Initially, two tests were performed to assess the sensitivity of various vascular and neural branches in the neck region, which served to determine whether to proceed with CM. The first test conducted was the vertebral artery sensitivity test (basilar artery) [37]. For this test, the participant was positioned in the supine position while the physical therapist held the participant’s head off the treatment table and performed a full backward tilt and cervical rotation to one side with the participant’s eyes open for 20 s. The test was considered positive if the participant reported headache, nausea, nystagmus, or any event related to hypoxia. Next, the Adson test was performed, in which the physical therapist carried out combined movements of extension, external rotation, and abduction in a stretched position while palpating the participant’s radial pulse [38]. Both tests were considered positive if the magnitude of the radial pulse decreased.

CM was performed using the Toggle Recoil technique, a specialized method characterized by a high-velocity and low-amplitude thrust applied manually between the C0 and C2 vertebrae [39], without any rotational movement. A physical therapist, with more than three years of clinical experience with this specific technique, was responsible for applying the CM technique to all participants. The procedure followed established biomechanical parameters (speed, amplitude, and segmental specificity) described in the literature. The occurrence of joint cavitation (audible or palpable) was considered an operational indicator confirming the effective delivery of the HVLA thrust. Importantly, if cavitation did not occur, the thrust was not repeated for safety reasons; however, the maneuver was still considered valid provided that all technical requirements of CM were respected. The participants were positioned on the treatment table in the lateral decubitus position, with the side to be treated facing upward (the technique was applied unilaterally). The physical therapist stood behind the participant and gently placed the pisiform area of one hand on the skin between the C0 and C2 vertebrae, while using the other hand to stabilize the wrist. A rapid triceps contraction was then performed by the physical therapist, resulting in a rapid elbow extension (toggle), immediately followed by a recoil (recoil). Randomization of the side to receive the CM condition followed a Latin square strategy, which ensured that the participant remained blinded to the treated side.

Manual Massage (MM): For the MM by myofascial release condition, a design adapted by Monteiro et al. [23] was used, but adapted to the muscles of the lower limbs. A physical therapist with more than eight years of clinical experience with this specific technique was responsible for applying the MM technique to all participants. The physical therapist performed a caudal to cranial MM release slip (palmar region) and applied, unilaterally, in random order, a single 120 s set to the anterior (i.e., quadriceps) and posterior (i.e., hamstrings) thigh, posterior (i.e., calf) leg, and trunk (i.e., multifidus) regions. Randomization of the segment to receive the MM condition followed a Latin square strategy, which ensured that the participant remained blinded to the treated side. Anterior thigh positions were performed lying in a supine position between the acetabulum and the quadriceps tendon. Posterior thigh and leg were performed in lying (prone) position between the ischial tuberosity and popliteal fossa, and Achilles tendon and popliteal fossa, respectively. The multifidus region was treated with the participant in a prone position, targeting the area between the sacral base and the proximal transitional zone of the spine, specifically between C7 and T1. The physical therapist would slide the fingers or hands over the skin in the target region while the participant remained lying and relaxed. MM was applied at different angles to target all areas with controlled pressure by a pain level scale score of 6 out of 10. Participants were instructed to maintain their usual respiratory pattern throughout all MM protocols. MM protocols were performed at the same time of day to avoid possible diurnal variations.

Simulated CM (SHAM): SHAM condition was replicated with the same setup and participant positioning as the CM condition. During this condition, the participants were positioned laterally on the treatment table with the side to be manipulated facing upward. The physical therapist stood behind the participant, gently placing the pisiform area of one hand on the skin over the C0 to C2 region and the other hand on the cervical drop mechanism of the table. The intentional disengagement of the cervical drop was then executed and lightly accompanied by the other hand, allowing the participants to hear the drop mechanism while being unaware of the absence of manipulation. However, instead of applying the high-velocity and low-amplitude thrust, the therapist performed only superficial manual contact and light passive movements within the neutral range of the cervical joint. No cavitation or end-range force was applied, ensuring the absence of therapeutic manipulation while maintaining participant blinding regarding the intervention received.

Control: The Control condition included no interventions (CM, MM, or Sham); only HRV and BP measurements were taken. Participants arrived at the laboratory and were positioned supine on the treatment table for 10 min to acclimatize to the temperature, lighting, and noise of the environment. Following this acclimatization period, baseline and “post-condition” values were collected, with an 8 min interval (equivalent to the duration of the MM intervention) between these two time points.

### 2.6. Outcome Measures

Blood Pressure: Systolic and diastolic BP were measured using an automatic oscillometric device (Omron Hem 7113, São Paulo, Brazil) [30]. Measurements were taken on the left arm, following the recommendations of the American Heart Association [40]. For optimal analysis and comparison of BP responses, the following time points were established: (a) Resting (Baseline); (b) immediately post-session (1st time point); (c) 15 min post-session (2nd time point); (d) 30 min post-session (3rd time point); (e) 45 min post-session (4th time point); (f) 60 min post-session (5th time point).

Heart Rate Variability (HRV): HRV recordings were performed with participants lying in the supine position. The Polar H10 heart rate monitor was connected to a chest strap, secured firmly yet comfortably around the thorax, with the sensor and electrode positioned at the lower third of the sternum (approximately over the xiphoid process). The heart rate monitor was paired via Bluetooth with the Elite HRV app installed on an iPhone [41,42]. Data were collected for 15 min at rest (Baseline) and up to 60 min (Post-0, Post-15, Post-30, Post-45, Post-60) after the experimental condition of the day.

Measurements were taken in a quiet room with participants in a supine position, minimal external noise, and temperature controlled between 20 and 22.8 °C. Raw iR-R data from the Elite HRV app were exported and subsequently analyzed using specialized software (Kubios, V.2.0, Kuopio, Finland). Time-domain and frequency-domain analyses were performed. Fast Fourier transform was used for frequency domain analysis. HRV parameters assessed in this study included: Root Mean Square of Successive Differences (RMSSDms), Standard Deviation of Normal-to-Normal intervals (SDNNms), Natural Logarithm of RMSSD (LNms), Percentage of NN50 (pNN50%), mean inter-beat interval (Mean iR-Rms), Total Power (ms^2^), Low-Frequency Power (LFms^2^), High-Frequency Power (HFms^2^), and the LF/HF Ratio. RMSSDms and pNN50% are considered sensitive markers of short-term parasympathetic modulation, whereas SDNNms reflects overall HRV by integrating both sympathetic and parasympathetic inputs. The logarithmic transformation of RMSSD (LNms) was applied to improve statistical robustness by reducing data skewness. Mean iR-Rms provides a measure of average cardiac cycle duration, while spectral indices (Total Power, LFnu, HFnu, and LF/HF ratio) describe the distribution of variance across frequency bands, thereby offering a more detailed characterization of autonomic modulation. Together, these indices provide relevant insights into the balance between sympathetic and parasympathetic activities of the autonomic nervous system [8], enabling a comprehensive analysis of cardiovascular autonomic regulation under the experimental conditions.

For optimal analysis and comparison of HRV responses, the following time points were established: (a) Resting (Baseline); (b) immediately post-session (1st time point); (c) 15 min post-session (2nd time point); (d) 30 min post-session (3rd time point); (e) 45 min post-session (4th time point); (f) 60 min post-session (5th time point).

### 2.7. Data Analysis

Fifteen healthy [43] women with non-elevated BP [36] were recruited based on an a priori sample size calculation (effect size = 0.50; 1 − β = 0.85; α = 0.05; nonsphericity correction = 1.0) [44], based values on BP highlighted by Monteiro et al. [23] study, using G*Power [45] indicated that fourteen participants would be adequate to achieve the statistical power.

The normality of the data distribution was assessed using the Shapiro–Wilk test and graphical inspection of histogram and QQ-plots. Normality was rejected for all variables examined. In this scenario, median and interquartile range were used to as measures of central tendency and dispersion, respectively, and inferential analyses were conducted using the Friedman Test [46]. The significance level was set at 5%. Adjusted pairwise comparisons were automatically conducted by the software in case of a statistically significant difference. When no statistically significant difference was found, Wilcoxon signed rank tests were conducted aiming to assess effect sizes. Carryover and period effects were controlled by Latin square randomization and predefined washout periods. Effect sizes were used to interpret within condition results according to the following Cohen’s guidelines for r: 0.1 (small effect), 0.3 (medium effect), and 0.5 (large effect) [46]. IBM SPSS Statistics 20 software was used for analysis (SPSS, Chicago, IL, USA).

## 3. Results

Figure 1 presents a flowchart showing the number of eligible participants excluded and the reason for their exclusion. All participants completed the study and provided data on all outcome measures.

The participants’ characteristics are described in Table 2.

The statistical results for carryover effects are summarized in Table 3, indicating that no significant time × order interactions (0.335–0.954) were observed across the analyzed outcomes.

### 3.1. Blood Pressure

No statistically significant differences were observed in Systolic BP across any of the time points (Table 4). The only statistically significant increase was observed in Diastolic BP within the Control condition (Post-45 vs. Baseline; *p* = 0.004) (Table 5) with large effect size (r = 0.66).

### 3.2. Heart Rate Variability—Time-Domain

Within condition evaluations of RMSSD_ms_ index significantly increased in the Control condition between Post-0 and Baseline (*p* = 0.032; r = 0.56) and between Post-15 and Baseline (*p* = 0.023; r = 0.57) (Table 6). Similarly, in the Sham condition, RMSSD_ms_ index significantly increased between Post-15 and Baseline (*p* = 0.014; r = 0.60) (Table 6). Lastly, in the CM condition, RMSSD_ms_ index significantly increased between Post-15 and Baseline (*p* = 0.027; r = 0.60) and between Post-60 and Baseline (*p* = 0.010; r = 0.17) (Table 6). A statistically significant difference was observed between CM and MM conditions at the Post-15 time point (*p* = 0.049); however, this difference was no longer significant following pairwise comparisons (Table 6).

The main findings demonstrated statistically significant within-condition increases in RMSSD_ms_ for Post-0 (d = 0.56) and Post-15 (d = 0.57), accompanied by a large effect size. However, while these changes reached statistical significance, they should not be directly interpreted as clinically meaningful without considering the magnitude of effect sizes and the broader physiological context. Significant increases in the SDNN_ms_ index were observed in the Control condition between Post-45 and Baseline (*p* = 0.037; r = 0.55) (Table 6). In the CM condition, significant increases were noted between Post-45 and Baseline (*p* = 0.014; r = 0.60) and between Post-60 and Baseline (*p* = 0.019; r = 0.58) (Table 6). Similarly, in the MM condition, increases were observed between Post-60 and Baseline (*p* = 0.019; r = 0.58) (Table 6). No statistically significant differences were found between conditions at any time point (Table 6). Regarding the analyses of pNN50% index, no statistically significant differences were observed between any time points in the Control, Sham, and MM conditions after pairwise comparisons. However, within condition analyses in the CM condition revealed statistically significant increases between Post-0 and Baseline (*p* = 0.044; r = 0.54), Post-15 and Baseline (*p* = 0.044; r = 0.54), and Post-60 and Baseline (*p* = 0.019; r = 0.58) (Table 6). No statistically significant differences between protocols were identified at any time point (Table 6).

Within-condition analyses of LN(RMSSD) index revealed statistically significant differences in all conditions except for the MM condition (Table 6). In the Control condition, significant increases were observed between Post-0 and Baseline (*p* = 0.027; r = 0.57). In the Sham condition, significant increases occurred between Post-15 and Baseline (*p* = 0.011; r = 0.61). For the CM condition, significant increases were identified between Post-15 and Baseline (*p* = 0.027; r = 0.57) and between Post-60 and Baseline (*p* = 0.010; r = 0.62) (Table 6). No statistically significant differences were found between protocols at any time point. For the mean iR-R, within-condition analyses revealed statistically significant differences across all conditions (Table 6). In the Control condition, differences were observed at all time points (*p* < 0.05; Table 6). In the Sham condition, significant increases were identified between Post-0 and Baseline (*p* = 0.027; r = 0.57) and between Post-15 and Baseline (*p* = 0.010; r = 0.62) (Table 6). For the CM condition, significant increases were observed at all investigated time points (*p* < 0.05; Table 6), except between Post-0 and Baseline (*p* = 0.051) (Table 6). Lastly, in the MM condition, significant increases were noted between Post-45 and Baseline (*p* < 0.001; r = 0.73) and between Post-60 and Baseline (*p* = 0.019; r = 0.58) (Table 6). No statistically significant differences were observed between protocols at any time point.

### 3.3. Heart Rate Variability—Frequency-Domain

Significant within-condition increases in Total Power ms^2^ index were observed in the Control, CM, and MM conditions (Table 7). In the Control condition, a significant difference was noted between Post-45 and Baseline (*p* = 0.027; r = 0.57). In the CM condition, differences were observed between Post-30 and Baseline (*p* = 0.037; r = 0.55) and between Post-60 and Baseline (*p* = 0.001; r = 0.74). In contrast, for the MM condition, the observed differences were diluted after pairwise comparisons. No statistically significant differences were found between protocols at any time point.

Analyses of the LF Power (ms^2^) index revealed a statistically significant difference in the CM condition between Post-60 and Baseline (*p* = 0.001; r = 0.71) (Table 6). No statistically significant differences were observed between protocols at any time point. Regarding the HF Power (ms^2^) index, a statistically significant difference was noted in the Sham condition, but this difference was diluted after pairwise comparisons (Table 6). Similarly, no statistically significant differences between protocols were found at any time point. For the LF/HF ratio, a within-condition statistically significant difference was identified in the MM condition between Post-60 and Baseline (*p* = 0.022; r = 0.55). Additionally, statistically significant differences were observed between the MM and Control conditions (*p* = 0.014) and between the MM and CM conditions (*p* = 0.043).

## 4. Discussion

Significant within-condition increases (*p* < 0.05) were observed in RMSSD_ms_, SDNN_ms_, PNN50%, LF Power (ms^2^), and LF/HF ratio, each with a large effect size. The results of the present study partially confirm the first hypothesis, which suggested that the CM condition would demonstrate superior sympatho-vagal modulation compared to the Control and Sham conditions. In the within-condition analysis, only the PNN50% index and LN Power (ms^2^) showed significant increases for CM, with no significant changes observed for Control and Sham. These results indicate an increase in both parasympathetic control (PNN50%) and sympathetic control (LF power (ms^2^)), but without predominance between them, which was reflected in the lack of significance in the LF/HF ratio. Even without active intervention, factors such as the placebo effect and natural adaptive physiological processes may influence physiological measurements due to non-specific effects [47,48,49]. However, these changes must be interpreted cautiously, as within-condition statistical differences do not necessarily indicate clinically meaningful adaptations. By combining inferential results with effect size analysis, we observed short-term trends suggestive of autonomic modulation, though without sufficient evidence to establish superiority of the experimental conditions over Control or Sham.

Nevertheless, Win et al. [21] investigated the effect of upper CM on sympatho-vagal modulation by HRV measurement and found different results from the present investigation. These authors observed an increase in HRV parameters, which may indicate a greater influx of vagal nerve activity. In addition, there was a decrease in LF_nu_ parameters (*p* = 0.01) and in the sympatho-vagal balance (HF/LF ratio), confirming the positive effects of upper CM on autonomic modulation. On the other hand, Picchiottino et al. [10] compared the effects of manipulation of the mid-thoracic region (T5 vertebra) and the sham effect on HRV responses. Picchiottino et al. [10] found an increase in the iR-R, as well as in the HF_nu_ (vagal pathway) and LFnu (vagal and sympathetic pathway) parameters, LF/HF ratio, RMSSD_ms_ (vagal pathway), and SDNN_ms_ (vagal and sympathetic pathway). The results by Picchiottino et al. [10] suggest that there is no relationship between T5 manipulation and autonomic modulation, as both parameters (parasympathetic and sympathetic) increased, thus reinforcing the mixed action of the vagus nerve and supporting the findings of the present study. The results described here suggest a possible relationship between manipulation and vagus nerve stimulation for predominantly parasympathetic input, regardless of the location (anatomical height) of this stimulus, a fact that was not predominantly observed in the results of the present study.

The second hypothesis proposes a higher magnitude of systolic BP reduction through the MM condition compared to the Control and Sham. This expectation was based on mechanisms described for foam rolling interventions, such as increased nitric oxide bioavailability [27] and enhanced arterial perfusion [28]. Contrary to this hypothesis, no significant reductions in systolic or diastolic BP were observed. The results of the present study do not support this hypothesis, as no significant reduction in BP values was observed. Contrary to the results of the present study, but without comparing experimental conditions to placebo, Liao et al. [50] conducted a systematic review with meta-analysis and found that massage techniques significantly contributed to reductions in systolic (−7.39 mmHg; effect size = −0.728) and diastolic (−5.04 mmHg; effect size = −0.334) BP. Regarding the results related to sliding MM in women, the literature only points to the study by Monteiro et al. [23], which aimed to evaluate this. In contrast to the results of the present study, Monteiro et al. [23] observed reduction in systolic BP at 50 (*p* = 0.011; r = 2.61; ∆ = −4.0 mmHg) and 60 (*p* = 0.011; r = −2.74; ∆ = −4.0 mmHg) minutes after application of MM as an experimental condition.

Similarly, but using foam rolling as an intervention strategy, Lastova et al. [9] observed a reduction in systolic BP, which differs from the findings of the present study. Lastova et al. [9] investigated BP response 10 and 30 min after a foam rolling session for the thigh (adductors, posterior, anterior, and lateral), calf (gastrocnemius), and back (upper and lower). The authors observed a significant decrease in systolic BP with a concomitant increase in vagal modulation up to 30 min after foam rolling. It is worth noting that Lastova et al. [9] measured BP only 30 min after foam rolling, leaving a gap in interpretation as to whether this reduction may persist beyond that time point. The specific physiological mechanisms underlying the BP response have been previously studied. The first to investigate potential effects related to BP control were Okamoto et al. [27], who observed a higher concentration of nitric oxide after foam rolling, indicating a greater vasodilatory effect that could reduce systolic BP and heart rate. Hotfiel et al. [28] observed increased local arterial perfusion in the lateral thigh region after foam rolling. The authors also associate these changes with vasodilation caused by increased nitric oxide after FR. Therefore, the fact that different techniques (e.g., MM and foam rolling) elicit the same central responses suggests that these effects may be attributed to therapeutic touch [51,52,53], which triggers responses at the level of the ANS. It is hypothesized that mechanoreceptors located in muscle and fascia, when activated, reduce muscle tone, thereby promoting an increase in parasympathetic response and the release of neuropeptides and endocannabinoids, leading to a subsequent reduction in blood pressure [17].

### 4.1. Limitation

An important consideration in interpreting the results of this study is the paced nature of the MM slide condition, both within and between individuals. This design feature can be viewed as both a limitation and a strength. On the one hand, the lack of strict control may reduce the internal validity of the results, as the number and duration of slides could influence the outcomes. On the other hand, allowing the slide to be freely paced increases ecological validity and generalizability and better reflects real-world rehabilitation scenarios. All participants in this study were women. Notably, resting cardiovascular parameters do not appear to vary significantly with different phases of the menstrual cycle. Interestingly, Queiroz et al. [54] highlighted those certain phases of the menstrual cycle, such as menstrual bleeding, may influence the magnitude of BP reduction but not its overall occurrence. Although caution is warranted in generalizing these findings, it is essential to emphasize that the primary aim of this study was not to investigate underlying mechanisms but rather to assess BP. We intentionally chose not to control for menstrual cycle phases in order to better represent real-world conditions, thereby enhancing the ecological validity of the findings.

### 4.2. Future Research Recommendation

Although the present study provides valuable insights into the effects of the intervention in apparently healthy women, the intentional and convenience-based sampling strategy inevitably limits the generalizability of the findings. The main differences between the sexes are primarily in the underlying physiological mechanisms rather than in the BP responses themselves [54]. To advance this field of investigation, future studies should consider expanding recruitment strategies to include both sexes and a broader age range, thereby enabling comparative analyses of potential sex-specific or age-related differences in cardiovascular and neuromuscular responses. Moreover, incorporating hypertensive populations or individuals with comorbidities would enhance the translational applicability of the results, providing a clearer picture of the clinical relevance of the intervention. Stratified analyses across different physiological and clinical profiles may help disentangle whether the observed effects are consistent or moderated by these variables.

Methodologically, future research would also benefit from multicenter trials with larger sample sizes, supported by formal a priori power analyses. Such designs would strengthen external validity and provide more robust evidence for clinical practice. Additionally, integrating complementary outcomes, such as autonomic modulation markers, advanced posturographic analysis, or longitudinal follow-up, could further elucidate the mechanisms underlying the observed responses.

## 5. Conclusions

Although no statistically significant between-condition differences were detected, within-condition changes with moderate-to-large effect sizes suggest potential clinical relevance of CM and MM. These preliminary findings align with ACSM recommendations that emphasize the importance of effect sizes and may indicate greater translational significance in populations with non-elevated BP. Therefore, caution will be warranted when interpreting and applying these results, as they reveal an emerging gap in the literature that warrants further investigation.

## Figures and Tables

**Figure 1 healthcare-13-02554-f001:**
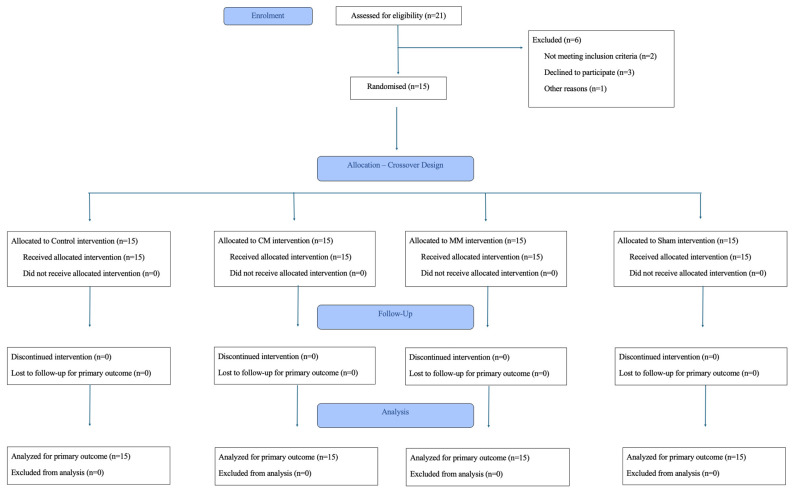
Flowchart.

**Table 1 healthcare-13-02554-t001:** Median values of systolic and diastolic blood pressure.

Resting Blood Pressure (mmHg)	Upper CM	Manual Massage	SHAM	Control
Systolic Blood Pressure	105	110	109	107
Diastolic Blood Pressure	70	68	75	70

CM = upper cervical manipulation; SHAM = simulated CM.

**Table 2 healthcare-13-02554-t002:** Sample’s characteristics expressed as mean and interquartile range.

	Dominant Limb	Age (Years)	Body Mass (kg)	Height (m)	BMI (kg/m^2^)	Physical Activity Level (min)
n = 15	R = 13; L = 2	21 (9)	57.30 (11.90)	1.65 (0.07)	21.99 (3.34)	290 (710)

R = right; L = left.

**Table 3 healthcare-13-02554-t003:** Carryover effect assessed by the time × intervention order interaction.

Diastolic Blood Pressure mmHg_lf	0.39	0.945
HF POWER ms2_lf	0.88	0.483
LF POWER ms2_lf	0.48	0.869
LF-HF RATIO _lf	1.20	0.335
LN (RMSSD) ms_lf	0.32	0.921
Mean R-R ms_lf	1.06	0.415
PNN50%_lf	0.47	0.856
RMSSD ms_lf	0.28	0.954
SDNN ms_lf	0.55	0.788
Systolic Blood Pressure mmHg_lf	0.39	0.946
TOTAL POWER ms2_lf	0.79	0.573

**Table 4 healthcare-13-02554-t004:** Baseline and post-intervention systolic blood pressure values with their respective effect sizes (ES).

Systolic Blood Pressure							
Protocol	Baseline	Post-0	Post-15 min	Post-30 min	Post-45 min	Post-60 min	Within Condition
Control	107 (13)	108 (14)	106 (10)	105 (12)	107 (11)	109 (11)	0.539
ES (r) (classification)		0.08 (no effect)	0.18 (small)	0.16 (small)	0.21 (small)	0.10 (small)	
Sham	109 (11)	109 (9)	107 (9)	108 (10)	107 (13)	107 (9)	0.504
ES (r) (classification)		0.06 (no effect)	0.25 (small)	0.13 (small)	0.14 (small)	0.19 (small)	
CM	105 (14)	106 (11)	112 (12)	111 (8)	108 (12)	108 (15)	0.160
ES (r) (classification)		0.16 (small)	0.21 (small)	0.14 (small)	0.11 (small)	0.01 (no effect)	
MM	110 (10)	108 (9)	109 (14)	108 (7)	108 (8)	108 (6)	0.221
ES (r) (classification)		0.21 (small)	0.04 (no effect)	0.22 (small)	0.15 (small)	0.22 (small)	

Sham: Simulated CM replicated by the same setup and participant positioning as the CM condition; CM = cervical manipulation; MM = manual massage.

**Table 5 healthcare-13-02554-t005:** Baseline and post-intervention diastolic blood pressure values with their respective effect sizes (ES).

Diastolic Blood Pressure							
Protocol	Baseline	Post-0	Post-15 min	Post-30 min	Post-45 min	Post-60 min	Within Condition
Control	70 (12)	71 (7)	70 (8)	72 (12)	74 (10) *	76 (6)	0.000 *
ES (r) (classification)		0.08 (no effect)	0.13 (small)	0.06 (no effect)	0.66 (large)	0.49 (medium)	
Sham	75 (11)	71 (17)	73 (9)	73 (7)	74 (11)	76 (9)	0.068
ES (r) (classification)		0.28 (small)	0.19 (small)	0.02 (no effect)	0.03 (no effect)	0.17 (small)	
CM	70 (7)	71 (7)	72 (6)	71 (8)	70 (9)	71 (8)	0.238
ES (r) (classification)		0.12 (small)	0.36 (medium)	0.24 (small)	0.14 (small)	0.25 (small)	
MM	68 (8)	71 (7)	73 (8)	72 (6)	71 (5)	72 (6)	0.225
ES (r) (classification)		0.26 (small)	0.40 (medium)	0.32 (medium)	0.20 (small)	0.26 (small)	

* level of significance set at 5%; Sham: Simulated CM replicated by the same setup and participant positioning as the CM condition; CM = cervical manipulation; MM = manual massage.

**Table 6 healthcare-13-02554-t006:** Baseline and post-intervention RMSSD, SDNN, LN (RMSSD), pNN50, and Mean R-R values with their respective effect sizes (ES).

		RMSSD_ms_					
	Baseline	Post-0 min	Post-15 min	Post-30 min	Post-45 min	Post-60 min	Within Condition
Control	40.05 (43.5)	63.39 (48.23)	61.52 (26.37)	49.30 (24.58)	63.49 (36.73)	52.23 (35.94)	0.008 *
ES (r) (classification)		0.56 (large) *	0.57 (large) *	0.16 (small)	0.39 (medium)	0.28 (small)	
Sham	41.88 (42.51)	79.44 (55.99)	71.10 (23.02)	60.53 (39.03)	75.76 (46.31)	54.08 (33.68)	0.007 *
ES (r) (classification)		0.51 (large)	0.60 (large) *	0.39 (medium)	0.49 (medium)	0.17 (small)	
CM	32.86 (18.58)	59.37 (49.07)	56.91 (34.55)	49.91 (45.99)	48.71 (52.16)	58.04 (52.27)	0.008 *
ES (r) (classification)		0.51 (large)	0.60 (large) *	0.39 (medium)	0.49 (medium)	0.17 (small) *	
MM	38.05 (33.86)	57.11 (40.27)	52.61 (30.84)	64.02 (33.74)	52.63 (31.04)	56.78 (32.42)	0.159
ES (r) (classification)		0.47 (medium)	0.29 (small)	0.32 (medium)	0.31 (medium)	0.31 (medium)	
		**SDNNms**					
Control	52.84 (35.68)	73.50 (57.76)	79.16 (29.03)	78.12 (29.79)	78.85 (25.45)	79.01 (23.18)	0.036 *
ES (r) (classification)		0.23 (small)	0.42 (medium)	0.42 (medium)	0.55 (large) *	0.44 (medium)	
Sham	62.08 (40.69)	74.55 (32.98)	77.42 (27.26)	79.53 (2348)	81.55 (25.88)	71.88 (36.25)	0.260
ES (r) (classification)		0.21 (small)	0.30 (medium)	0.37 (medium)	0.28 (small)	0.34 (medium)	
CM	47.19 (27.95)	62.64 (34.83)	86.43 (27.46)	76.26 (31.27)	93.21 (26.07)	79.01 (45.66)	0.008 *
ES (r) (classification)		0.24 (small)	0.46 (medium)	0.39 (medium)	0.60 (large) *	0.58 (large) *	
MM	54.03 (30.39)	67.39 (19.49)	69.79 (26.40)	78.49 (29.63)	71.23 (32.15)	79.01 (29.03)	0.027 *
ES (r) (classification)		0.23 (small)	0.30 (medium)	0.37 (medium)	0.48 (medium)	0.58 (large)	
		**pNN50%**					
Control	14.00 (37)	46.00 (41)	33.00 (32)	30.00 (33)	37.00 (38)	28.00 (30)	0.034 *
ES (r) (classification)		0.52 (large)	0.42 (medium)	0.10 (small)	0.27 (small)	0.16 (small)	
Sham	24.00 (38)	54.00 (47)	47.00 (25)	40.00 (24)	46.00 (27)	40.00 (38)	0.019 *
ES (r) (classification)		0.53 (large)	0.51 (large)	0.41 (medium)	0.46 (medium)	0.20 (small)	
CM	9.00 (24)	41.00 (39)	32.00 (30)	30.00 (32)	29.00 (36)	35.00 (32)	0.013 *
ES (r) (classification)		0.54 (large) *	0.54 (large) *	0.39 (medium)	0.44 (medium)	0.58 (large) *	
MM	17.00 (40)	34.00 (41)	32.00 (43)	31.00 (32)	34.00 (29)	32.00 (27)	0.040 *
ES (r) (classification)		0.32 (medium)	0.50 (large)	0.49 (medium)	0.43 (medium)	0.43 (medium)	
		**LN (RMSSD) (ms)**					
Control	3.69 (0.93)	4.25 (0.82)	4.12 (0.52)	3.90 (0.47)	4.15 (0.63)	3.96 (0.64)	0.020 *
ES (r) (classification)		0.57 (large) *	0.49 (medium)	0.16 (small)	0.35 (medium)	0.28 (small)	
Sham	3.73 (0.91)	4.37 (0.88)	4.26 (0.31)	4.10 (0.58)	4.33 (0.72)	3.99 (0.65)	0.006 *
ES (r) (classification)		0.51 (large)	0.61 (large) *	0.40 (medium)	0.49 (medium)	0.16 (small)	
CM	3.49 (0.54)	4.08 (0.78)	4.04 (0.52)	3.91 (0.82)	3.89 (0.90)	4.06 (0.85)	0.008 *
ES (r) (classification)		0.53 (large)	0.57 (large) *	0.49 (medium)	0.49 (medium)	0.62 (large) *	
MM	3.64 (0.80)	4.04 (0.70)	3.96 (0.57)	4.16 (0.64)	3.96 (0.54)	4.04 (0.57)	0.139
ES (r) (classification)		0.48 (medium)	0.35 (medium)	0.35 (medium)	0.34 (medium)	0.35 (medium)	
		**MEAN iR-R (ms)**					
Control	760.27 (93.89)	832.38 (141.16)	864.56 (105.99)	844.58 (64.55)	875.01 (113.64)	833.73 (187.20)	0.002 *
ES (r) (classification)		0.55 (large) *	0.66 (large) *	0.58 (large) *	0.69 (large) *	0.55 (large) *	
Sham	830.15 (198.40)	919.70 (187.45)	897.00 (229.02)	866.99 (214.50)	918.14 (179.70)	833.73 (204.33)	0.008 *
ES (r) (classification)		0.57 (large) *	0.62 (large) *	0.44 (medium)	0.48 (medium)	0.28 (small)	
CM	764.71 (159.71)	857.01 (112.75)	859.38 (131.51)	853.64 (76.45)	846.67 (123.12)	913.03 (133.61)	0.003 *
ES (r) (classification)		0.53 (large)	0.60 (large) *	0.58 (large) *	0.55 (large) *	0.66 (large) *	
MM	779.00 (87.89)	838.71 (134.60)	861.39 (116.44)	829.93 (167.58)	860.01 (128.53)	866.62 (116.20)	0.001 *
ES (r) (classification)		0.26 (small)	0.51 (large)	0.41 (medium)	0.73 (large) *	0.58 (large) *	

* level of significance set at 5%; Sham: Simulated CM replicated by the same setup and participant positioning as the CM condition; CM = cervical manipulation; MM = manual massage.

**Table 7 healthcare-13-02554-t007:** Baseline and post-intervention Total Power, LF Power, HF Power, and LF/HF Ratio values with their respective effect sizes (ES).

		TOTAL POWER ms^2^					
	Baseline	Post-0 min	Post-15 min	Post-30 min	Post-45 min	Post-60 min	Within Condition
Control	1346.67 (2004.78)	2464.36 (4455.47)	2471.27 (2486.22)	3302.12 (2981.87)	3039.01 (2798.19)	3107.56 (2753.03)	0.033 *
ES (r) (classification)		0.42 (medium)	0.35 (medium)	0.23 (small)	0.57 (large) *	0.17 (small)	
Sham	1968.19 (2919.92)	2381.00 (3796.07)	3111.46 (1751.67)	3021.98 (1302.74)	3334.23 (5174.11)	2605.74 (2370.88)	0.497
ES (r) (classification)		0.34 (medium)	0.33 (medium)	0.31 (medium)	0.31 (medium)	0.22 (small)	
CM	944.18 (704.78)	2464.36 (3450.03)	2439.50 (1410.69)	3021.98 (3887.12)	2381.87 (4353.62)	3242.34 (5241.07)	0.001 *
ES (r) (classification)		0.27 (small)	0.31 (medium)	0.55 (large) *	0.51 (large)	0.74 (large) *	
MM	1844.65 (1933.13)	2160.07 (3516.33)	2339.49 (2221.31)	3236.94 (3368.09)	2320.10 (2588.11)	3087.02 (1161.28)	0.049 *
ES (r) (classification)		0.07 (small)	0.21 (small)	0.42 (medium)	0.39 (medium)	0.44 (medium)	
		**LF POWER ms^2^**					
Control	650.67 (1237.53)	1467.53 (2658.45)	1540.16 (1464.43)	1539.34 (1543.21)	1795.87 (1438.67)	1900.21 (1514.26)	0.115
ES (r) (classification)		0.27 (small)	0.37 (medium)	0.31 (medium)	0.38 (medium)	0.29 (small)	
Sham	927.42 (1570.22)	1081.43 (1200.03)	1540.16 (653.86)	1554.75 (1032.19)	1795.87 (1428.13)	1282.71 (1741.09)	0.529
ES (r) (classification)		0.08 (no effect)	0.22 (small)	0.17 (small)	0.29 (small)	0.30 (medium)	
CM	544.98 (281.41)	1050.85 (2108.26)	1402.23 (834.30)	1359.34 (1595.44)	1896.11 (1917.92)	2027.90 (3582.68)	0.001 *
ES (r) (classification)		0.10 (small)	0.35 (medium)	0.41 (medium)	0.49 (medium)	0.71 (large) *	
MM	677.05 (1051.34)	800.48 (648.48)	1415.73 (1146.53)	1359.34 (1230.96)	1479.18 (594.68)	1632.17 (965.79)	0.115
ES (r) (classification)		0.01 (no effect)	0.14 (small)	0.14 (small)	0.20 (small)	0.25 (small)	
		**HF POWER ms^2^**					
Control	693.30 (782.99)	1907.67 (2228.37)	1201.11 (2288.40)	1174.41 (1585.22)	1734.03 (1695.55)	784.84 (1655.61)	0.091
ES (r) (classification)		0.31 (medium)	0.21 (small)	0.14 (small)	0.21 (small)	0.05 (no effect)	
Sham	971.18 (1469.83)	1152.90 (2539.24)	1792.10 (1614.62)	1734.03 (885.14)	1734.03 (2310.83)	1026.63 (1422.45)	0.15
ES (r) (classification)		0.37 (medium)	0.53 (large)	0.40 (medium)	0.47 (medium)	0.08 (no effect)	
CM	381.95 (530.48)	1311.49 (1728.06)	1247.38 (977.78)	843.58 (1808.54)	679.32 (2591.78)	1271.04 (2771.01)	0.073
ES (r) (classification)		0.48 (medium)	0.36 (medium)	0.38 (medium)	0.29 (small)	0.41 (medium)	
MM	778.91 (1426.57)	1331.16 (2020.04)	1345.73 (675.12)	1726.65 (1992.47)	1109.43 (2271.16)	1271.04 (1486.11)	0.130
ES (r) (classification)		0.41 (medium)	0.24 (small)	0.32 (medium)	0.22 (small)	0.23 (small)	
		**LF/HF RATIO**					
Control	0.98 (0.83)	1.00 (0.37)	1.28 (1.29)	1.14 (1.93)	1.10 (2.22)	1.87 (2.74)	0.111
ES (r) (classification)		0.04 (no effect)	0.12 (small)	0.28 (small)	0.18 (small)	0.42 (medium)	
Sham	1.19 (1.09)	0.82 (0.76)	0.89 (0.76)	0.93 (0.53)	0.86 (0.81)	1.87 (2.01)	0.149
ES (r) (classification)		0.23 (small)	0.35 (medium)	0.36 (medium)	0.13 (small)	0.17 (small)	
CM	1.34 (0.91)	0.94 (1.18)	1.05 (1.17)	1.17 (1.08)	1.23 (1.51)	1.12 (1.87)	0.98
ES (r) (classification)		0.27 (small)	0.02 (no effect)	0.05 (no effect)	0.14 (small)	0.17 (small)	
MM	1.38 (1.18)	0.69 (0.67)	0.78 (1.09)	0.80 (0.55)	0.94 (0.97)	1.49 (2.14)	0.022 *
ES (r) (classification)		0.30 (medium)	0.10 (small)	0.01 (no effect)	0.24 (small)	0.55 (large) *	

* level of significance set at 5%; CM= cervical manipulation; MM = manual massage.

## Data Availability

The data presented in this study are available as Appendix A to this submission and can be openly accessed on the journal’s website. All Appendix A accompanying this publication are intended to ensure full data transparency.

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
