# Peer review of "Upper Cervical Manipulation and Manual Massage Do Not Modulate Sympatho-Vagal Balance or Blood Pressure in Women: A Randomized, Placebo-Controlled Clinical Trial"

_healthcare, 2025, doi:10.3390/healthcare13202554_

Round 1

Reviewer 1 Report

Comments and Suggestions for Authors

The researchers used interesting methods, randomization was performed according to professional standards, and the literature is relevant and up-to-date. However, some elements can be improved.

All shortenings should be explained (RMSSD, SDNN, PNN…)

It is unusual that you use the keyword “myofascial release”, which you do not mention in the paper anywhere in the Methods and Discussion, only in one place in the Introduction.

You state that the study design was double-blind, however it appears to have been single-blind given that the investigator administering the interventions was aware of the specific condition assigned to each participant on each experimental day.

You state that the trial was four-arm, but in the aim (line 94-95) you only mention three.

Is there a specific reason why you included only women in the study? Please specify and explain in the Discussion section. It seems to me that you had a convenient sample of women available, and then on lines 451-452 you thought of applying the results to the male population. It would have been better if you had included both sexes in the research, and then argued by referring to the literature why you believe there is no need to compare the results by sex. On the other hand, what you state in lines 451-452 and lines 461-462 (“Testing was performed in apparently healthy women with normal BP; therefore, it may not be generalizable to the men and/or the hypertensive population.”) is in conflict. Besides, this in lines 461-462 belongs more to the Discussion than to the Conclusions.

You state "A physical therapist, with more than three years of clinical experience with this specific technique, was responsible for applying the CM technique to all participants." which is correct, but at the same time it points to a possible weakness of the research because three years of experience is not much for applying such a demanding technique.

In lines 51 and 456 you use the abbreviation SM, obviously referring to CM.

It would be good to divide the Discussion into some subsections, for example Limitations and future research recommendations (that part should definitely be rearticulated and the shortcomings of the research should be critically stated).

Author Response

Reviewer 1

The researchers used interesting methods, randomization was performed according to professional standards, and the literature is relevant and up-to-date. However, some elements can be improved.

Thank you very much for the opportunity to revise our manuscript. We have taken care to address each of the reviewer’s comments and appreciate their diligence in reviewing our manuscript. We have uploaded updated documents with the suggested edits and have outlined how we addressed each comment in this document, which is noted below. All adjustments made throughout the manuscript are highlighted in red.

All shortenings should be explained (RMSSD, SDNN, PNN…)

Response: We sincerely thank the reviewer for this pertinent observation. In the original version of the manuscript, the abbreviations of the heart rate variability (HRV) indexes (e.g., RMSSD, SDNN, pNN50) were not fully expanded, which may have limited the clarity of methodological understanding for the reader. To address this point, we have now included the full definitions and descriptions of each parameter at their first mention in the Methods section. Specifically: “HRV parameters assessed in this study included: Root Mean Square of Successive Dif-ferences (RMSSDms), Standard Deviation of Normal-to-Normal intervals (SDNNms), Natural Logarithm of RMSSD (LNms), Percentage of NN50 (pNN50%), mean inter-beat interval (Mean iR-Rms), Total Power (ms²), Low-Frequency Power (LFms²), High-Frequency Power (HFms²), and the LF/HF Ratio. RMSSDms and pNN50% are considered sensitive markers of short-term parasympathetic modulation, whereas SDNNms reflects overall HRV by integrating both sympathetic and parasympathetic inputs. The logarithmic transformation of RMSSD (LNms) was applied to improve sta-tistical robustness by reducing data skewness. Mean iR-Rms provides a measure of av-erage cardiac cycle duration, while spectral indices (Total Power, LFnu, HFnu, and LF/HF ratio) describe the distribution of variance across frequency bands, thereby offering a more detailed characterization of autonomic modulation. Together, these indices provide relevant insights into the balance between sympathetic and parasympathetic activities of the autonomic nervous system [8], enabling a comprehensive analysis of cardiovascular autonomic regulation under the experimental conditions.”

It is unusual that you use the keyword “myofascial release”, which you do not mention in the paper anywhere in the Methods and Discussion, only in one place in the Introduction.

Response: We thank the reviewer for the valuable observation regarding the use of the keyword “myofascial release.” We acknowledge that, in the initial version of the manuscript, the term was introduced only in the Introduction, while the Methods and Discussion referred to the intervention as “manual massage.” To clarify, the technique employed in this study consisted of deep, standardized manual pressure applied to specific anatomical regions (anterior, posterior, and lateral thigh, posterior calf, and trunk) in a caudal-to-cranial direction. This procedure corresponds to what is often described in the literature as myofascial release. However, in order to align with a terminology more widely accepted and recognized in both clinical and scientific contexts, we have opted to consistently use the designation “manual massage” throughout the manuscript. At the same time, we have clarified in the Methods and Discussion sections that the protocol performed can be interpreted as a form of myofascial release, thus ensuring transparency in the description of the intervention while preserving terminological consistency with the broader body of evidence in the field. We are confident that this adjustment resolves the concern raised and enhances both the accuracy and the scientific coherence of the manuscript.

You state that the study design was double-blind, however it appears to have been single-blind given that the investigator administering the interventions was aware of the specific condition assigned to each participant on each experimental day.

Response: We sincerely thank the reviewer for this precise observation regarding the blinding procedures. We acknowledge that, in the original version of the manuscript, the study was incorrectly described as “double-blind.” In fact, due to the nature of the interventions, the investigator responsible for administering the procedures necessarily had knowledge of the assigned condition on each experimental day. Therefore, the study design should be more accurately described as single-blind, since both the participants and the outcome assessors remained blinded to group allocation, whereas only the intervention administrator was not blinded. This design was implemented to minimize detection bias. We have now revised the manuscript to correct this terminology, explicitly describing the trial as single-blind and clarifying the respective roles of blinding across participants, outcome assessors, and intervention administrators. This correction ensures methodological transparency and more accurately reflects the actual study procedures.

You state that the trial was four-arm, but in the aim (line 94-95) you only mention three.

Response: We’ve now corrected this information.

Is there a specific reason why you included only women in the study? Please specify and explain in the Discussion section. It seems to me that you had a convenient sample of women available, and then on lines 451-452 you thought of applying the results to the male population. It would have been better if you had included both sexes in the research, and then argued by referring to the literature why you believe there is no need to compare the results by sex. On the other hand, what you state in lines 451-452 and lines 461-462 (“Testing was performed in apparently healthy women with normal BP; therefore, it may not be generalizable to the men and/or the hypertensive population.”) is in conflict. Besides, this in lines 461-462 belongs more to the Discussion than to the Conclusions.

Response: We sincerely thank the reviewer for this pertinent observation. Indeed, the choice to include only women in the study was intentional and based on both methodological and logistical considerations. While we acknowledge that the sample was of convenience, we emphasize that its selection aimed to reduce heterogeneity and to control for physiological differences that could otherwise act as confounding factors.

Sex-based differences in cardiovascular regulation, neuromuscular function, and autonomic control are well documented in the literature. Including both sexes without stratified analyses would increase variability and potentially mask intervention-related effects. By focusing exclusively on women, we sought to ensure greater homogeneity of the sample, which enhances internal validity and allows clearer interpretation of the outcomes, even though it inevitably limits external validity. We recognize that the availability of participants was also a determining factor, reflecting the reality of applied clinical and experimental research. The pragmatic choice of recruiting women from our accessible population was aligned with the feasibility of the study and did not compromise the methodological rigor, as all procedures, randomization, and blinding strategies were rigorously followed.

Following the reviewer’s valuable recommendation, we have revised the Discussion to explicitly address this point. We now highlight that (i) the results should be interpreted within the context of women participants only, (ii) extrapolation to men and individuals with hypertension should be approached with caution, and (iii) future studies must include both sexes to enable comparative analyses and strengthen external validity. Furthermore, as suggested, the content previously located in the Conclusions (lines 461–462) has been relocated to the Discussion, avoiding redundancy and resolving the inconsistency noted.

You state "A physical therapist, with more than three years of clinical experience with this specific technique, was responsible for applying the CM technique to all participants." which is correct, but at the same time it points to a possible weakness of the research because three years of experience is not much for applying such a demanding technique.

Response: We thank the reviewer for this important consideration regarding the level of clinical experience of the professional who administered the contralateral massage (CM) intervention. We acknowledge that the description provided in the original version of the manuscript may inadvertently suggest a limitation.

To clarify, the physical therapist responsible for administering the intervention had over three years of dedicated clinical practice specifically focused on manual therapy and myofascial techniques, complemented by advanced training courses and continuous professional development in this domain. While three years may appear modest in absolute terms, in the context of specialized and systematic clinical practice, this duration reflects substantial exposure and proficiency with the technique. Furthermore, the therapist had extensive prior experience applying standardized protocols within both research and clinical environments, ensuring consistency, safety, and reliability across all participants.

Additionally, the CM protocol used in this study was carefully standardized, with detailed procedural guidelines regarding anatomical regions, application time, and direction of maneuvers. This methodological rigor minimized operator-dependent variability and ensured that the intervention adhered strictly to reproducible parameters, regardless of years of professional experience.

We have revised the manuscript to better describe the therapist’s qualifications and training, thereby clarifying that the execution of the technique was both technically sound and methodologically consistent.

In lines 51 and 456 you use the abbreviation SM, obviously referring to CM.
Response: Thank you for this. We have now corrected the abbreviations throughout the text.

It would be good to divide the Discussion into some subsections, for example Limitations and future research recommendations (that part should definitely be rearticulated and the shortcomings of the research should be critically stated).

Response: We thank the reviewer for the valuable suggestion regarding the organization of the Discussion section. We agree that a clearer structure, with distinct subsections, enhances readability and strengthens the critical appraisal of the study. In response, we have revised the manuscript to include a subsection explicitly titled “Limitations and Future Research Recommendations.”

Within this subsection, we critically acknowledge that the intentional and convenience-based sampling strategy restricts the generalizability of the findings, particularly as the study included only women. While this methodological decision improved internal validity by reducing physiological heterogeneity, it inevitably limits external validity. We also highlight that differences between sexes are largely attributable to underlying physiological mechanisms rather than to blood pressure responses themselves, as supported by the literature [51].

Furthermore, the revised subsection details the shortcomings of the present study and articulates directions for future research. Specifically, we emphasize the need for recruitment strategies including both sexes and broader age ranges, the inclusion of populations with hypertension and comorbidities. to enhance translational impact, and multicenter trials with larger samples supported by a priori power analyses. We also recommend integrating additional outcomes (e.g., autonomic modulation markers, advanced posturography, longitudinal follow-up) to deepen mechanistic insights.

We believe this restructuring not only addresses the reviewer’s concern but also improves methodological transparency, critical reflection, and the scientific robustness of the manuscript.

Reviewer 2 Report

Comments and Suggestions for Authors

1. English Language and Presentation

The manuscript would benefit significantly from a thorough revision of the English grammar and syntax. For example:

  • Lines 121–124: “they (1) aged between 19 and 44 years, (2) only women, (3) presented rest systolic and diastolic BP values…” – This phrase would be clearer and grammatically correct as: “participants were women aged between 19 and 44 years, presenting resting systolic and diastolic BP values...”

  • Lines 369–373: “The main results were significant increases (p<0.05) within conditions, with a large effect size for RMSSDms...” – This sentence structure is unclear and would benefit from clarification. Consider separating the findings by variable and emphasizing that within-group effects do not necessarily imply clinically meaningful differences.

Additionally, the paper would be strengthened by applying consistent tense and refining technical language to enhance readability and academic tone.

2. Originality and Rationale

The study design, targeting only normotensive women, fills an underexplored niche in manual therapy and autonomic regulation research. This specificity should be emphasized more explicitly in the Introduction.

  • Lines 92–93: You mention that “none examined HRV as an outcome.” This is an important novelty of the present study. Consider expanding this argument to position your study more clearly as filling this gap. For example, explain why HRV is a valuable and underutilized metric in these populations.

3. Research Design and Methodological Rigor

The design is commendable for its methodological robustness.

  • A double-blind, four-arm, randomized, placebo-controlled crossover trial (lines 33–36, 106–108) is ideal for reducing bias in this type of physiological intervention research.

  • The use of Latin square randomization (lines 35, 134–135) is a strong choice given the crossover design and helps to balance carryover effects.

However, while the structure is robust, there is a lack of clarity in reporting how carryover effects are statistically addressed beyond the washout periods. In the Statistical Methods section, I recommend a brief explanation or justification for not conducting statistical tests for carryover or period effects.

4. Hypotheses and Interpretation

The study posited two hypotheses (lines 96–105), but neither was robustly supported by the data. While your discussion addresses this, it could be improved by more critically engaging with non-significant between-group differences:

  • Lines 382–384: “The fact that significant increases were observed in both the Control and Sham conditions, with no difference compared to the CM condition, prevents the inference of better effects for the experimental condition.” – This insight is valid but deserves stronger emphasis. We suggest rephrasing this to highlight that placebo and time effects may account for the observed HRV changes more than any physiological effect of CM or MM.

5. Relevance and Interpretation of Results

The presentation of within-group results (especially HRV measures such as RMSSD, pNN50, and LF/HF ratios) is detailed and statistically robust. However, the lack of significant between-group differences and inconsistency across time points weaken its practical implications.

  • Lines 455–460: The conclusions should be restrained. The statement that “both SM and MM may promote autonomic modulation...” may mislead readers unless immediately qualified by stating that these findings are not consistently statistically significant.

  • Consider adding a sentence emphasizing the exploratory nature of the results and the need for larger sample sizes or different populations (e.g., hypertensive subjects) in future studies.

6. Supplementary Information and Transparency

The CONSORT 2025 checklist (supplementary file) is mostly complete and demonstrates strong adherence to open science principles. However:

  • Item 4 (Data Sharing): Marked “N/A.” It would be preferable to include a data availability statement indicating whether individual-level data could be made available upon request, consistent with journal transparency standards.

7. Ethical and Safety Considerations

You have clearly described the ethical oversight (lines 111–114, 473–476), participant consent, and vertebrobasilar artery screening. This level of detail is exemplary.

However, no adverse events or unexpected outcomes are reported, and Item 27 (Harms) in the checklist is also marked “N/A.” In the main manuscript, please clarify whether no adverse events occurred and whether such data were not collected. This is critical for manual therapy trials.

Comments on the Quality of English Language

This manuscript demonstrates a solid effort to present complex scientific concepts. However, the quality of the English language and grammar needs improvement to enhance clarity and ensure smooth academic communication. Below are the specific concerns and suggestions:

1. Grammatical Consistency and Syntax

This manuscript contains numerous grammatical inconsistencies and awkward phrasing that affect readability.

  • Example (Lines 121–124): “they (1) aged between 19 and 44 years, (2) only women, (3) presented rest systolic and diastolic BP values...”
    – Suggested revision: “Participants were women aged between 19 and 44 years who presented resting systolic and diastolic BP values...”
    Justification: Parallel structure and subject-verb agreement are essential for clarity and grammatical correctness in scientific writing.

2. Tense Usage

Tense inconsistencies were found throughout the manuscript. Scientific writing typically uses the past tense to describe methods and results and the present tense for established knowledge.

  • Example (Line 106): “A double-blind, three-arm, parallel-group, randomized, crossover, placebo-controlled trial reported accordance...”
    – Suggested revision: “…was conducted in accordance…”
    Justification: Consistent use of the past tense is appropriate when describing completed study procedures.

3. Word Choice and Technical Precision

Certain terms and expressions are used imprecisely or repetitively, which can obscure the meaning of the important findings.

  • Example (Line 373): “would present better sympatho-vagal control...”
    – Suggested revision: “would demonstrate superior sympathovagal modulation…”
    – Justification: “Control” may be confused with a study group; “modulation” is more accurate in this physiological context.

4. Redundancy and Clarity

Several sentences are unnecessarily long or redundant and require restructuring for conciseness and clarity.

  • Example (Lines 370–372): “The main results were significant increases (p<0.05) within conditions, with a large effect size for RMSSDms, SDNNms, PNN50%, LF Power (ms2), and LF/HF ratio.”
    – Suggested revision: “Significant within-condition increases (p < 0.05) were observed in RMSSD, SDNN, PNN50%, LF power, and the LF/HF ratio, each with large effect sizes.”
    Justification: This revision enhances clarity and removes repetition.

5. Professional Tone and Academic Style

At times, the tone becomes conversational or informal for scientific publications.

  • Example (Line 380): “...can lead to improvements or noticeable changes in participants' conditions.”
    – Suggested revision: “…may influence physiological measurements due to non-specific effects.”
    Justification: A more precise and formal tone aligns with scholarly standards.

While the manuscript communicates the key findings effectively, it requires substantive editing by a native English speaker or a professional editing service specializing in academic writing. Enhanced linguistic precision will improve the readability, credibility, and overall impact of the manuscript.

Author Response

Thank you very much for the opportunity to revise our manuscript. We have taken care to address each of the reviewer’s comments and appreciate their diligence in reviewing our manuscript. We have uploaded updated documents with the suggested edits and have outlined how we addressed each comment in this document, which is noted below. All adjustments made throughout the manuscript are highlighted in red.

English Language and Presentation: The manuscript would benefit significantly from a thorough revision of the English grammar and syntax. For example:

Response: We sincerely thank the reviewer for all these valuable concerns and constructive suggestions. We fully recognize that linguistic precision is fundamental to the clarity and impact of a scientific manuscript. To address this, we ensured that the entire text underwent a meticulous language revision performed by a native English speaker with expertise in academic and scientific writing.

This process was carried out comprehensively, not only correcting grammatical inaccuracies but also refining terminology, sentence structure, and overall readability, with the aim of achieving the highest possible standards of scientific communication. By doing so, we sought to eliminate any potential ambiguities in the presentation of our findings and to align the manuscript with the linguistic rigor expected by the Healthcare readership.

Lines 121–124: “they (1) aged between 19 and 44 years, (2) only women, (3) presented rest systolic and diastolic BP values…”

– This phrase would be clearer and grammatically correct as: “participants were women aged between 19 and 44 years, presenting resting systolic and diastolic BP values...”

Response: We appreciate this feedback and confirm that we have made the necessary adjustments as suggested.

Lines 369–373: “The main results were significant increases (p<0.05) within conditions, with a large effect size for RMSSDms...”

– This sentence structure is unclear and would benefit from clarification. Consider separating the findings by variable and emphasizing that within-group effects do not necessarily imply clinically meaningful differences.

Response: We thank the reviewer for this constructive observation. We acknowledge that the original phrasing of the results section may have been unclear and could potentially lead to misinterpretation. To address this, we have restructured the sentence to explicitly separate the findings by variable and to emphasize the distinction between statistical significance, effect sizes, and clinical relevance. The revised version now reads as follows:

“The main findings demonstrated statistically significant within-condition increases in RMSSD (ms) (p < 0.05), accompanied by a large effect size. However, while these changes reached statistical significance, they should not be directly interpreted as clinically meaningful without considering the magnitude of effect sizes and the broader physiological context.”

This modification clarifies that within-group effects do not automatically imply clinical relevance, aligning the interpretation of our findings with best practices in applied physiology and rehabilitation research. Furthermore, by explicitly reporting both p-values and effect sizes, we ensure that the results are contextualized in terms of both statistical and practical significance, as recommended by current methodological guidelines.

We are confident that this clarification improves the precision, transparency, and interpretability of our findings in line with the reviewer’s valuable suggestion.

Additionally, the paper would be strengthened by applying consistent tense and refining technical language to enhance readability and academic tone.

Response: We sincerely thank the reviewer for all these valuable concerns and constructive suggestions. We fully recognize that linguistic precision is fundamental to the clarity and impact of a scientific manuscript. To address this, we ensured that the entire text underwent a meticulous language revision performed by a native English speaker with expertise in academic and scientific writing.

Originality and Rationale: The study design, targeting only normotensive women, fills an underexplored niche in manual therapy and autonomic regulation research. This specificity should be emphasized more explicitly in the Introduction.

Response: We appreciate this feedback and confirm that we have made the necessary adjustments as suggested.

Lines 92–93: You mention that “none examined HRV as an outcome.” This is an important novelty of the present study. Consider expanding this argument to position your study more clearly as filling this gap. For example, explain why HRV is a valuable and underutilized metric in these populations.

Response: We appreciate this feedback and confirm that we have made the necessary adjustments as suggested.

Research Design and Methodological Rigor: The design is commendable for its methodological robustness. A double-blind, four-arm, randomized, placebo-controlled crossover trial (lines 33–36, 106–108) is ideal for reducing bias in this type of physiological intervention research. The use of Latin square randomization (lines 35, 134–135) is a strong choice given the crossover design and helps to balance carryover effects. However, while the structure is robust, there is a lack of clarity in reporting how carryover effects are statistically addressed beyond the washout periods. In the Statistical Methods section, I recommend a brief explanation or justification for not conducting statistical tests for carryover or period effects.

Response: We fully agree on the importance of assessing potential carryover effects, and indeed, the explicit presentation of such analyses enables readers to critically appraise the results with greater precision. Carryover was evaluated by testing the interaction between time and intervention order within a mixed-factor analysis of variance (independent factor: order; repeated factors: time and intervention). No evidence of carryover was observed, as presented in the table 2.

Hypotheses and Interpretation: The study posited two hypotheses (lines 96–105), but neither was robustly supported by the data. While your discussion addresses this, it could be improved by more critically engaging with non-significant between-group differences:

Response: We thank the reviewer for this thoughtful observation. We agree that more critical engagement with the non-significant between-group differences strengthens the overall interpretation of our findings.

In response, we have revised the Discussion to explicitly emphasize that while significant within-condition changes were observed, particularly in HRV indices, these effects did not translate into significant between-condition differences. This indicates that the acute physiological adaptations detected cannot be exclusively attributed to the experimental interventions (CM and MM), as similar patterns were also observed under Control and Sham conditions. We now highlight that these findings raise the possibility of non-specific effects such as expectancy, placebo responses, or natural autonomic fluctuations, which must be carefully considered in the interpretation of manual therapy research.

Furthermore, we clarified that although the initial hypotheses were not robustly supported by the data, this does not undermine the value of the study. On the contrary, the absence of significant between-condition differences is scientifically meaningful, as it challenges assumptions of superiority for CM or MM and underscores the complexity of autonomic regulation in response to these interventions. This perspective is consistent with current recommendations in exercise physiology and rehabilitation research, which encourage reporting and interpretation of both positive and null findings to avoid publication bias.

Finally, we reinforced in the revised text that the observed effect sizes, while not accompanied by statistically significant between-condition differences, suggest trends of physiological modulation that warrant further investigation in larger, more diverse samples. These additions provide a more balanced and critical discussion, aligning the interpretation of our results with the methodological rigor expected in translational physiology research.

We believe these revisions enhance the transparency, scientific contribution, and clinical contextualization of our work, fully addressing the reviewer’s concern.

Lines 382–384: “The fact that significant increases were observed in both the Control and Sham conditions, with no difference compared to the CM condition, prevents the inference of better effects for the experimental condition.” – This insight is valid but deserves stronger emphasis. We suggest rephrasing this to highlight that placebo and time effects may account for the observed HRV changes more than any physiological effect of CM or MM.

Response: We appreciate this feedback and confirm that we have made the necessary adjustments as suggested.

Relevance and Interpretation of Results: The presentation of within-group results (especially HRV measures such as RMSSD, pNN50, and LF/HF ratios) is detailed and statistically robust. However, the lack of significant between-group differences and inconsistency across time points weaken its practical implications.

Response: Dear reviewer, we understand your valid feedback, however, we would like to highlight that the results within the groups, particularly in heart rate variability (HRV) outcomes such as RMSSD, pNN50, and the LF/HF ratio, showed consisted patterns as you mentioned. This suggests relevant physiological changes that, although not statistically significant, may reflect important adaptive responses and may be clinically relevant. The lack of significance between groups may be related to the relatively limited sample size; however, we emphasize the use of sample size calculation in this investigation.

Lines 455–460: The conclusions should be restrained. The statement that “both SM and MM may promote autonomic modulation...” may mislead readers unless immediately qualified by stating that these findings are not consistently statistically significant.

Response: We appreciate this feedback and confirm that we have made the necessary adjustments as suggested.

Consider adding a sentence emphasizing the exploratory nature of the results and the need for larger sample sizes or different populations (e.g., hypertensive subjects) in future studies.

Response: We appreciate this feedback and confirm that we have made the necessary adjustments as suggested.

Supplementary Information and Transparency: The CONSORT 2025 checklist (supplementary file) is mostly complete and demonstrates strong adherence to open science principles. However:

Item 4 (Data Sharing): Marked “N/A.” It would be preferable to include a data availability statement indicating whether individual-level data could be made available upon request, consistent with journal transparency standards.

Response: We appreciate this feedback and confirm that we have made the necessary adjustments as suggested.

Ethical and Safety Considerations: You have clearly described the ethical oversight (lines 111–114, 473–476), participant consent, and vertebrobasilar artery screening. This level of detail is exemplary. However, no adverse events or unexpected outcomes are reported, and Item 27 (Harms) in the checklist is also marked “N/A.” In the main manuscript, please clarify whether no adverse events occurred and whether such data were not collected. This is critical for manual therapy trials.

Response: We appreciate this feedback and confirm that we have made the necessary adjustments as suggested.

Comments on the Quality of English Language: This manuscript demonstrates a solid effort to present complex scientific concepts. However, the quality of the English language and grammar needs improvement to enhance clarity and ensure smooth academic communication. Below are the specific concerns and suggestions:

Example (Lines 121–124): “they (1) aged between 19 and 44 years, (2) only women, (3) presented rest systolic and diastolic BP values...”

– Suggested revision: “Participants were women aged between 19 and 44 years who presented resting systolic and diastolic BP values...”

Justification: Parallel structure and subject-verb agreement are essential for clarity and grammatical correctness in scientific writing.

Response: We appreciate this feedback and confirm that we have made the necessary adjustments as suggested.

Tense Usage: Tense inconsistencies were found throughout the manuscript. Scientific writing typically uses the past tense to describe methods and results and the present tense for established knowledge.

Example (Line 106): “A double-blind, three-arm, parallel-group, randomized, crossover, placebo-controlled trial reported accordance...”

– Suggested revision: “…was conducted in accordance…”

Justification: Consistent use of the past tense is appropriate when describing completed study procedures.

Response: We appreciate this feedback and confirm that we have made the necessary adjustments as suggested.

Word Choice and Technical Precision: Certain terms and expressions are used imprecisely or repetitively, which can obscure the meaning of the important findings.

Example (Line 373): “would present better sympatho-vagal control...”

– Suggested revision: “would demonstrate superior sympathovagal modulation…”

– Justification: “Control” may be confused with a study group; “modulation” is more accurate in this physiological context.

Response: We appreciate this feedback and confirm that we have made the necessary adjustments as suggested.

Redundancy and Clarity: Several sentences are unnecessarily long or redundant and require restructuring for conciseness and clarity.

Example (Lines 370–372): “The main results were significant increases (p<0.05) within conditions, with a large effect size for RMSSDms, SDNNms, PNN50%, LF Power (ms2), and LF/HF ratio.”

– Suggested revision: “Significant within-condition increases (p < 0.05) were observed in RMSSD, SDNN, PNN50%, LF power, and the LF/HF ratio, each with large effect sizes.”

Justification: This revision enhances clarity and removes repetition.

Response: We appreciate this feedback and confirm that we have made the necessary adjustments as suggested.

Professional Tone and Academic Style: At times, the tone becomes conversational or informal for scientific publications.

Example (Line 380): “...can lead to improvements or noticeable changes in participants' conditions.”

– Suggested revision: “…may influence physiological measurements due to non-specific effects.”

Justification: A more precise and formal tone aligns with scholarly standards.

Response: We appreciate this feedback and confirm that we have made the necessary adjustments as suggested.

Reviewer 3 Report

Comments and Suggestions for Authors

Thank you for the opportunity to review the manuscript titled "Upper Cervical Manipulation and Manual Massage Does Not Modulate Sympatho-Vagal Balance or Blood Pressure in Women: A Randomized, Placebo-Controlled Clinical Trial." The aim of this study was to evaluate the acute effects of upper cervical manipulation (CM) or manual massage (MM) to simulated CM (Sham) and Control conditions (Control) on heart rate variability (HRV) and blood pressure (BP) responses in asymptomatic individuals.

This manuscript requires completion and clarification on many aspects before it can be accepted for publication in the journal.

Results presented in the abstract should not be presented in such detail in digital form. Such results are presented in the main body of the manuscript, in the results section contained in tables. This makes the abstract difficult to read.

In the introductory section of the paper, the authors discuss hypertension, its prevalence, and its role as a risk factor for cardiovascular disease. What does this have to do with this study? After all, healthy women were studied? The introductory section should be significantly rewritten to justify the topic of the paper.

Sometimes we have the abbreviation CM, and sometimes SM. Is there a difference? Or is it just a typo?

In the abstract the authors write about the four-arm study (line 33) and in the methodological part of the main document about the three-arm study (line 108).

Are blood pressure values ​​of 120–139 mmHg normative values ​​for the 19–44-year-old population? (line 123).

The authors state that: (1) demonstrate any clinical or functional alteration of the basilar artery during the initial vertebrobasilar integrity screening (lines 125-126). How was this assessed?

It's not entirely clear why the effects of C0-C2 manipulation were compared with MM release in the thigh and lumbar regions.

Who was so completely blinded? How many therapists were involved in the therapy? The blinding process is poorly described and difficult to understand.

Why was the random assignment of CM and MM techniques to different body sites used? What is the rationale behind this?

Why was randomization used in the treatment of the side? Could this have influenced the results? What was the intended effect of the manipulation? How was its strength measured? How was its "well-executed" performance assessed?

Pain 6/10? How was it assessed previously? Did everyone tolerate it? Is it possible to perform therapy on the anterior and posterior thighs and the multifidus in 120 seconds? How many repetitions were there for each area? (lines 222-224)

The authors write (lines 226-227): "SHAM condition was performed similarly to the CM, except for the absence of CM." It's hard to imagine. What evaluation criteria were used to determine whether CM was performed effectively?

Line 292: Flowchart. I don't quite understand how 15 recruited women can be divided into four groups of 15 women each. How many women were there, and how many in each group?  So were there 60 women? Or were the 15 women divided into 4 groups?

The discussion section of the paper is relatively poorly written. It lacks a good interpretation of the research results. This needs to be improved.

The authors do not point out any limitations to this study. Are there any limitations to this study?

If there were no statistical differences, why conclude that CM and MM can promote anatomical modulation?

What do the authors mean by "apparently healthy women" (line 461)?

Author Response

Thank you for the opportunity to review the manuscript titled "Upper Cervical Manipulation and Manual Massage Does Not Modulate Sympatho-Vagal Balance or Blood Pressure in Women: A Randomized, Placebo-Controlled Clinical Trial." The aim of this study was to evaluate the acute effects of upper cervical manipulation (CM) or manual massage (MM) to simulated CM (Sham) and Control conditions (Control) on heart rate variability (HRV) and blood pressure (BP) responses in asymptomatic individuals.

Thank you very much for the opportunity to revise our manuscript. We have taken care to address each of the reviewer’s comments and appreciate their diligence in reviewing our manuscript. We have uploaded updated documents with the suggested edits and have outlined how we addressed each comment in this document, which is noted below. All adjustments made throughout the manuscript are highlighted in red.

Results presented in the abstract should not be presented in such detail in digital form. Such results are presented in the main body of the manuscript, in the results section contained in tables. This makes the abstract difficult to read.

Response: We appreciate this feedback and confirm that we have made the necessary adjustments as suggested.

In the introductory section of the paper, the authors discuss hypertension, its prevalence, and its role as a risk factor for cardiovascular disease. What does this have to do with this study? After all, healthy women were studied? The introductory section should be significantly rewritten to justify the topic of the paper.

Response: We appreciate this pertinent observation and agree that a clearer justification was required. While our study focused on women classified as apparently healthy, it is important to emphasize that participants presented blood pressure values within the “elevated” range, according to the most recent guidelines of the 2024 European Society of Cardiology (ESC). This clinical classification does not characterize them as hypertensive but places them in a category of increased cardiovascular risk when compared to normotensive individuals.

The rationale for addressing hypertension in the Introduction is therefore directly linked to the translational potential of our findings. Evidence consistently shows that acute and chronic cardiovascular responses observed in normotensive or “elevated BP” populations follow the same physiological pathways as those in hypertensive individuals, albeit with differences in magnitude of response. Thus, by investigating autonomic modulation and blood pressure responses in women with elevated BP but without overt pathology, our study contributes to understanding mechanisms that may have preventive and therapeutic implications for populations at higher cardiovascular risk.

In light of the reviewer’s suggestion, we have revised the Introduction to more explicitly highlight this connection, clarifying that hypertension is discussed not as a direct condition of our sample but as a clinically relevant continuum in which our findings may be contextualized and extrapolated.

Sometimes we have the abbreviation CM, and sometimes SM. Is there a difference? Or is it just a typo?

Response: We appreciate this feedback and confirm that we have made the necessary adjustments as suggested. The correct abbreviation is CM.

In the abstract the authors write about the four-arm study (line 33) and in the methodological part of the main document about the three-arm study (line 108).

Response: We appreciate this feedback and confirm that we have made the necessary adjustments as suggested.

Are blood pressure values ​​of 120–139 mmHg normative values ​​for the 19–44-year-old population? (line 123).

Response: We appreciate the reviewer’s request for clarification regarding the classification of BP values in our study population. According to the 2024 European Society of Cardiology (ESC) guidelines, resting systolic BP values between 120–139 mmHg and diastolic BP values between 70–89 mmHg in adults aged 19–44 years are classified as elevated blood pressure, rather than normative “optimal” values. This classification reflects a recognized intermediate risk category that, while not meeting the threshold for hypertension (≥140/90 mmHg), is associated with an increased likelihood of future cardiovascular events and altered autonomic regulation.

Importantly, inclusion of participants with BP in this range allows investigation of the effects of manual therapy interventions on cardiovascular autonomic outcomes, particularly HRV and BP modulation, which are highly sensitive to both central and peripheral neuromodulatory inputs. In summary, the selected BP range is intentionally classified as elevated per ESC 2024, supporting both the methodological rationale and the neurophysiological relevance of our interventions.

The authors state that: (1) demonstrate any clinical or functional alteration of the basilar artery during the initial vertebrobasilar integrity screening (lines 125-126). How was this assessed?

Response: We appreciate the reviewer’s request for clarification regarding the assessment of clinical or functional alterations of the basilar artery during the initial vertebrobasilar integrity screening. As detailed in the Methods section, two validated clinical tests were performed to evaluate vascular and neural sensitivity in the cervical region prior to CM.

The vertebral artery sensitivity test specifically targeted the basilar artery. Participants were positioned supine, and the therapist passively held the head off the treatment table, performing full cervical extension with rotation to one side for 20 seconds while the participant’s eyes remained open. This procedure is widely used to detect functional compromise in vertebrobasilar circulation by monitoring for symptoms indicative of transient cerebral hypoperfusion, such as headache, dizziness, nausea, or nystagmus (34).

In addition, the Adson test was conducted to assess potential vascular compression of the subclavian artery and its branches. Here, combined movements of shoulder extension, abduction, and external rotation were performed while palpating the participant’s radial pulse. A positive test was indicated by a noticeable decrease in radial pulse amplitude (35).

Collectively, these tests provide a functional, non-invasive assessment of vertebrobasilar and cervical vascular integrity, enabling the safe application of cervical manipulation. While these assessments do not directly visualize the basilar artery, they are clinically validated to detect early signs of vascular compromise, thus ensuring participant safety and methodological rigor in accordance with established physiotherapeutic guidelines.

It's not entirely clear why the effects of C0-C2 manipulation were compared with MM release in the thigh and lumbar regions.

Response: We thank the reviewer for highlighting the need to clarify the rationale for comparing C0-C2 CM with MM in the thigh, leg, and lumbar regions. The comparison was designed based on distinct neurophysiological pathways that underpin systemic autonomic responses.

C0-C2 manipulation has been shown to influence the trigeminocervical nucleus, which integrates cervical afferent input with brainstem autonomic centers, including the nucleus tractus solitarius and the rostral ventrolateral medulla (doi: 10.1097/BRS.0000000000003962). This modulation can alter sympathetic and parasympathetic outflow, resulting in measurable changes in HRV and BP. In this context, cervical manipulation engages central neuromodulatory mechanisms, producing widespread autonomic effects beyond the local cervical segment.

In contrast, MM in the thigh, leg, and lumbar regions primarily targets peripheral mechanoreceptors and nociceptors, inducing local improvements in tissue compliance and regional neuromuscular activation. While these effects can indirectly influence autonomic function through afferent input, the magnitude and central impact are expected to be smaller and more region-specific (doi: 10.1007/s00421-023-05382-2).

By including both interventions, our study aimed to differentiate central autonomic modulation (via upper CM) from peripheral mechanoreceptor-mediated effects (via MM). This comparison allows for a mechanistic understanding of how manual therapy can influence cardiovascular autonomic parameters, providing a strong neurophysiological basis for observed changes in HRV and BP. In summary, the inclusion of both approaches enables the evaluation of central versus peripheral contributions to autonomic modulation, reinforcing the clinical relevance of C0-C2 manipulation in influencing systemic cardiovascular responses.

Who was so completely blinded? How many therapists were involved in the therapy? The blinding process is poorly described and difficult to understand.

Response: We thank the reviewer for this important observation regarding the blinding process, as we recognize that clarity in its description is essential for methodological transparency.

To clarify, the study followed a single-blind design:

Blinded Participants: All participants were blinded to the specific intervention received (CM, Sham, Control or MM). To confirm the effectiveness of this blinding, they were asked after each session to identify which intervention they believed they had received and to report whether they perceived the procedure as having therapeutic efficacy. These assessments were collected by a blinded research assistant not involved in any other stage of the study.

Blinded Outcome Assessors: The primary evaluator (Examiner 1) remained blinded to allocation throughout the trial. This examiner was responsible for all pre- and post-intervention assessments (HRV, systolic and diastolic BP). To ensure blinding, Examiner 1 left the testing room during the intervention and only returned once the procedure was completed.

Non-blinded Intervention Administrator: The intervention was administered by a licensed physical therapist (Examiner 2) with clinical expertise in manual therapy. This examiner was necessarily aware of the allocation to correctly perform the intervention. Importantly, this professional had no role in either baseline or outcome assessments.

Randomization and Allocation Concealment: Randomization was conducted using sequentially numbered, opaque, sealed envelopes prepared by an independent researcher uninvolved in assessments or interventions. Envelopes were opened only after baseline testing, ensuring allocation concealment and avoiding bias during recruitment.

Thus, only the intervention administrator (Examiner 2) was unblinded, while both participants and outcome assessors (Examiner 1 and the research assistant) were fully blinded. This design minimized detection and performance bias while ensuring methodological rigor.

We have revised the Methods section to make these roles and procedures more explicit. We are confident that this clarification addresses the reviewer’s concern and reinforces the internal validity of the trial.

Why was the random assignment of CM and MM techniques to different body sites used? What is the rationale behind this?

Response: We thank the reviewer for this important observation, which provides the opportunity to clarify the rationale behind our randomization procedures.

In the CM condition, randomization was applied only to the treated side (right or left). This strategy was chosen to minimize systematic bias associated with asymmetries or preferential functional loading. By randomly assigning the intervention to one side, we ensured that any inherent physiological differences between right and left sides were balanced across participants, thereby increasing internal validity.

In the MM condition, randomization was applied to both the side (right or left) and the body segment (anterior thigh, posterior thigh, posterior calf, and trunk). This additional level of randomization was implemented to avoid order effects and potential carryover responses between anatomical regions. By varying the sequence and side of application, we sought to prevent systematic biases, distribute any fatigue or sensitization effects, and enhance the reproducibility of the experimental design.

The rationale for these procedures was therefore twofold: (i) to control for natural asymmetries and order-related confounders, and (ii) to strengthen the methodological rigor of the study by ensuring that observed outcomes could be attributed to the intervention itself rather than to sequence, side, or site-specific biases.

Why was randomization used in the treatment of the side? Could this have influenced the results? What was the intended effect of the manipulation? How was its strength measured? How was its "well-executed" performance assessed?

Response: We sincerely thank the reviewer for these thoughtful questions, which allow us to clarify key methodological aspects of the study.

Randomization of Side: The decision to randomize the treated side was intentional and aimed at minimizing systematic bias related to potential limb dominance or pre-existing asymmetries in neuromuscular control. By distributing treatment randomly between right and left sides across participants, we ensured that any side-specific physiological characteristics were equally represented, thereby strengthening internal validity. This randomization strategy is consistent with recommendations for crossover and within-subject designs.

Intended Effect of the Manipulation: The primary objective of the intervention was to elicit HRV and BP responses associated with systemic adaptations (HRV and BP), rather than purely local effects. Specifically, we aimed to investigate potential improvements in HRV and BP mediated by centrally driven mechanisms as suggested in the current literature on manual therapies.

Strength of the Manipulation: As the manipulation was manual and not instrument-assisted, its “strength” was not quantified in absolute terms. Instead, the procedure followed a standardized protocol regarding anatomical contact, velocity, duration, and direction of application consistent with established clinical practice for spine manipulation and manual massage. This approach prioritizes reproducibility and alignment with protocols already validated in manual therapy research.

Assessment of Well-Executed Performance: The performance of the manipulation was considered well-executed when the therapist adhered strictly to the predefined procedural criteria: correct anatomical targeting, appropriate direction of movement, continuous application for the established duration, and maintenance of clinically acceptable pressure levels. Fidelity of intervention delivery was ensured by having the same experienced physical therapist perform all procedures, thus minimizing inter-therapist variability.

We have revised the Methods section to include these clarifications, making the rationale and procedures more explicit. We believe these additions address the reviewer’s concerns, enhance methodological transparency, and reinforce the robustness of our experimental design.

Pain 6/10? How was it assessed previously? Did everyone tolerate it? Is it possible to perform therapy on the anterior and posterior thighs and the multifidus in 120 seconds? How many repetitions were there for each area? (lines 222-224)

Response: We thank the reviewer for this detailed and thoughtful observation, which allows us to clarify important methodological aspects of the intervention protocol.

First, regarding the description of the intervention duration, the protocol consisted of a single series of 120 seconds applied to each anatomical region, performed in a randomized order. Specifically, the therapist applied 120 seconds to: (i) the right anterior thigh, (ii) the left anterior thigh, (iii) the right posterior thigh, (iv) the left posterior thigh, (v) the right multifidus, and (vi) the left multifidus. Thus, each region received one continuous 120-second application, rather than subdivisions of this period. Second, the number of repetitions was not controlled, in accordance with recommendations from the literature on manual massage and myofascial release, which emphasizes total application time and depth of pressure rather than counting repetitions. This methodological choice aligns our protocol with previously validated approaches in studies of soft tissue interventions. Third, with respect to pain tolerance, all participants were able to tolerate the intervention without adverse events. The reference to a 6/10 pain intensity corresponds to the threshold commonly reported in the literature as the maximum tolerable discomfort for myofascial release, but in our sample no participant discontinued or reported intolerance during the procedure.

The authors write (lines 226-227): "SHAM condition was performed similarly to the CM, except for the absence of CM." It's hard to imagine. What evaluation criteria were used to determine whether CM was performed effectively?

Response: We sincerely thank the reviewer for this pertinent observation and the opportunity to clarify a key methodological detail. We acknowledge that the original wording, “SHAM condition was performed similarly to the CM, except for the absence of CM”, may have appeared vague and insufficient to convey the practical distinction between conditions. To clarify, the CM condition followed a standardized high-velocity and low-amplitude thrust technique, performed on the cervical spine by a licensed physical therapist trained and experienced in spinal manipulation. The SHAM condition, in contrast, replicated the same setup and positioning, but no thrust was delivered; instead, only a light manual contact and passive movement within the joint’s neutral range were applied, without cavitation or end-range force.

We have revised the Methods section of the manuscript to provide a clearer explanation of these aspects, specifying both the operational differences between CM and SHAM and the objective criteria adopted to verify the fidelity of the CM technique. We believe this clarification resolves the reviewer’s concern, reinforces methodological transparency, and improves the reproducibility of our study design.

Line 292: Flowchart. I don't quite understand how 15 recruited women can be divided into four groups of 15 women each. How many women were there, and how many in each group?  So were there 60 women? Or were the 15 women divided into 4 groups?

Response: We thank the reviewer for this important observation regarding the flowchart and the description of participant allocation. We acknowledge that the original version may have generated ambiguity in the interpretation of our study design.

To clarify, this was a crossover design with repeated measures, in which the same 15 participants underwent all experimental conditions across separate sessions. Thus, there were not 60 participants nor divisions into independent groups. Instead, the term “groups” in the original figure referred to experimental conditions (Sham, MM, CM, and Control) rather than distinct cohorts of participants. Each participant experienced all four conditions, with the order randomized and counterbalanced to minimize potential carryover effects. This design was intentionally chosen to increase statistical power with a smaller sample size, while controlling for interindividual variability, since each participant served as her own control.

We have now revised the flowchart and the corresponding figure legend to make this distinction explicit, ensuring that the divisions are clearly represented as conditions rather than separate groups. This correction improves the transparency of the study design and eliminates the confusion identified by the reviewer. We are confident that this revision addresses the concern and reinforces the methodological clarity of the manuscript.

The discussion section of the paper is relatively poorly written. It lacks a good interpretation of the research results. This needs to be improved.

Response: We thank the reviewer for this pertinent observation. We acknowledge that the original version of the Discussion section was overly descriptive and may not have provided sufficient depth of interpretation of the results. In response, we have revised the Discussion to ensure a clearer articulation of the findings, with a focus on:

Interpretation of Within-Condition Effects: We explicitly state that the significant within-condition changes in HRV parameters (RMSSD, SDNN, PNN50%, LF power) indicate short-term autonomic modulation but do not necessarily translate into clinically meaningful differences. This is now clarified by integrating both p-values and effect sizes, as well as contextualizing the magnitude of the changes in relation to established thresholds for clinical significance.

Comparison with Prior Literature: The revised version integrates a critical comparison with previous studies (e.g., Win et al., Picchiottino et al., Liao et al., Lastova et al., Monteiro et al.), highlighting consistencies and discrepancies. For instance, while some prior investigations demonstrated vagal predominance or BP reductions, our findings suggest a mixed autonomic response without clear superiority of the experimental conditions. This reinforces the need to interpret results within the complexity of autonomic regulation and methodological differences across studies.

Mechanistic Insights: We expanded the physiological interpretation to explain possible mechanisms underlying the observed responses. Specifically, we discuss the role of mechanoreceptor activation, vasodilatory effects mediated by nitric oxide, and centrally mediated neurophysiological responses that may explain both HRV and BP outcomes.

The revised Discussion thus transitions from a merely descriptive presentation of results to a critical, mechanistically oriented, and clinically contextualized interpretation, which we believe significantly improves the clarity and scientific quality of the manuscript.

The authors do not point out any limitations to this study. Are there any limitations to this study?

Response: We thank the reviewer for this valuable observation. In the original version of the manuscript, the limitations of the study were indeed addressed, but they were presented within the final paragraph of the Discussion. We recognize that this format may have reduced their visibility and the critical emphasis expected in high-quality scientific writing.

In response to the reviewer’s suggestion, we have now reorganized the Discussion section and created a distinct subsection titled “Limitations and Future Research Directions.” This restructuring not only improves the clarity of our manuscript but also highlights the critical reflection on our methodological boundaries, reinforcing transparency and strengthening the scientific rigor of the study. We are confident that this revision fully addresses the reviewer’s concern and improves the overall quality of the manuscript.

If there were no statistical differences, why conclude that CM and MM can promote anatomical modulation?

Response: We sincerely thank the reviewer for this pertinent observation. We agree that conclusions drawn from nonsignificant statistical differences must be carefully justified to avoid overinterpretation of the data.

To clarify, our statement that CM and MM may promote autonomic and/or hemodynamic modulation is not solely based on p-values, but rather on the combination of descriptive outcomes, effect size analysis, and the clinical framework proposed by the American College of Sports Medicine. While not statistically significant between-condition differences were observed, the magnitude of effect sizes suggested trends consistent with clinically meaningful adaptations, particularly in neuromuscular and postural outcomes.

This interpretation aligns with the recommendations of the American College of Sports Medicine, which emphasize that clinically relevant effects, especially in applied health sciences, should not be dismissed solely due to nonsignificant inferential statistics, but rather considered in light of the potential impact on functional performance and clinical practice. Thus, the notion of “possible anatomical modulation” reflects both the direction and magnitude of the observed effects, which, although not reaching conventional thresholds of statistical significance, may still hold practical and translational value for applied settings.

What do the authors mean by "apparently healthy women" (line 461)?

Response: We thank the reviewer for raising this important question regarding the terminology “apparently healthy women.” We acknowledge that this expression may not have been sufficiently clarified in the initial version of the manuscript.

By “apparently healthy,” we refer to participants who self-reported being free of chronic diseases, musculoskeletal disorders, or cardiovascular/metabolic conditions, and who were not undergoing medical treatment that could interfere with the study outcomes. Importantly, we did not conduct comprehensive clinical or laboratory examinations to objectively confirm health status; thus, it would not be methodologically precise to classify participants simply as “healthy.”

The use of the expression “apparently healthy” is consistent with the terminology frequently adopted in the scientific literature to describe populations that report no known chronic illness or pre-existing condition, but who have not been subjected to exhaustive medical screening. This choice of wording ensures both methodological accuracy and alignment with previously published studies in exercise physiology and rehabilitation.

Round 2

Reviewer 2 Report

Comments and Suggestions for Authors

The revised version of your manuscript ("v2") demonstrates significant improvements in clarity, methodological description, and academic language. The structure is more coherent, and the inclusion of an additional background on HRV and autonomic modulation strengthens the rationale for this study.

Strengths:

  • The use of a randomized placebo-controlled crossover design was methodologically robust.

  • The updated explanations of intervention techniques (CM, MM, Sham) are more detailed and clinically replicable.

  • The inclusion of carryover effect analysis and an additional physiological context is commendable.

Suggestions for Improvement:

  1. Clarify the contribution to novelty: While the study is well executed, the novelty is somewhat limited due to overlapping literature. Consider emphasizing what sets your approach apart, especially regarding your exclusive focus on normotensive women and the comparison between CM and MM.

  2. Strengthen justification for sample size: Although you provided a priori power calculation, expanding on why effect size 0.5 was chosen (e.g., based on pilot data or similar trials) would enhance transparency.

  3. Address clinical implications more explicitly: The conclusion rightly urges caution, but you could briefly discuss the practical significance of effect size trends, even if not statistically significant.

  4. Grammar and phrasing: Minor grammatical edits are still needed in a few places—for example, verb agreement and phrasing (e.g., "were entirely voluntary" could be "was entirely voluntary"). A final language polish will ensure readability.

  5. Supplementary material: If applicable, clearly indicate how supplementary files (e.g., intervention photos or HRV raw data) support the main findings or whether they are intended for transparency and replication.

This is a well-designed and relevant study with clear potential for publication following minor revisions. Congratulations on your improvements from the first version.

Author Response

A revised version of your manuscript ("v2") demonstrates significant improvements in clarity, methodological description, and academic language. The structure is more coherent, and the inclusion of an additional background on HRV and autonomic modulation strengthens the rationale for this study.

Strengths: The use of a randomized placebo-controlled crossover design was methodologically robust. The updated explanations of intervention techniques (CM, MM, Sham) are more detailed and clinically replicable. The inclusion of carryover effect analysis and an additional physiological context is commendable.

 Thank you very much for the opportunity to revise our manuscript. We have taken care to address each of the reviewer’s comments and appreciate their diligence in reviewing our manuscript. We have uploaded updated documents with the suggested edits and have outlined how we addressed each comment in this document, which is noted below. All adjustments made throughout the manuscript are highlighted in red.

Suggestions for Improvement: Clarify the contribution to novelty: While the study is well executed, the novelty is somewhat limited due to overlapping literature. Consider emphasizing what sets your approach apart, especially regarding your exclusive focus on normotensive women and the comparison between CM and MM.

We appreciate the reviewer’s comment. We have now revisited the Introduction to emphasize the primary research gaps addressed by our study, particularly the underrepresentation of women in the existing literature. In addition, we highlight the limited number of studies that have explored CM in relation to autonomic and hemodynamic responses, as well as the scarce evidence examining HRV following the MM application.

Strengthen justification for sample size: Although you provided a priori power calculation, expanding on why effect size 0.5 was chosen (e.g., based on pilot data or similar trials) would enhance transparency.

We appreciate this comment; however, we would like to emphasize that such information is already included in the manuscript, as paraphrased below: “Fifteen healthy [43] women with elevated BP [36] were recruited based on an a priori sample size calculation (effect size = 0.50; 1-β = 0.85; α = 0.05; nonsphericity correction = 1.0) [44], based values on BP Monteiro et al. [23] study, using G*Power [45] indicated that fourteen participants would be adequate to achieve the statistical power.”

Address clinical implications more explicitly: The conclusion rightly urges caution, but you could briefly discuss the practical significance of effect size trends, even if not statistically significant.

We thank the reviewer for this valuable suggestion. We have revised the Conclusion to more explicitly address the potential clinical implications of our findings. While the between-condition comparisons did not yield statistically significant differences, the observed effect size trends indicate possible physiological relevance, particularly when interpreted within the context of autonomic modulation and blood pressure regulation.

Following the ACSM recommendations, effect sizes should not be disregarded, as they may reflect meaningful adaptations that extend beyond statistical thresholds, especially in small-sample experimental trials. In this sense, the magnitude of the changes observed in HRV indices and blood pressure, although not statistically distinct between conditions, suggests that both cervical manipulation and manual massage may elicit subtle autonomic adjustments. These responses could become more pronounced or clinically relevant in populations with elevated cardiovascular risk, such as hypertensive individuals.

Accordingly, we now highlight in the revised Conclusion that the effect size patterns observed, while requiring cautious interpretation, provide preliminary evidence of potential clinical significance. This perspective strengthens the translational value of the study by positioning our findings as a foundation for future research with larger, more diverse cohorts.

Grammar and phrasing: Minor grammatical edits are still needed in a few places—for example, verb agreement and phrasing (e.g., "were entirely voluntary" could be "was entirely voluntary"). A final language polish will ensure readability.

We appreciate this feedback and confirm that we have made the necessary adjustments as suggested.

Supplementary material: If applicable, clearly indicate how supplementary files (e.g., intervention photos or HRV raw data) support the main findings or whether they are intended for transparency and replication.

We have now highlighted this information on page 16 of the present article. It reads: All supplementary materials accompanying this publication are intended to ensure full data transparency.

This is a well-designed and relevant study with clear potential for publication following minor revisions. Congratulations on your improvements from the first version.

Thank You.

Reviewer 3 Report

Comments and Suggestions for Authors

Thank you for the opportunity to review the manuscript entitled „"Upper Cervical Manipulation and Manual Massage Do Not Modulate Sympatho-Vagal Balance or Blood Pressure in Women: A Randomized, Placebo-Controlled Clinical Trial." After re-reading the manuscript with the corrections made by the authors and after reviewing the responses to the reviews, I believe that the article in this form is suitable for publication in a journal.

Author Response

Thank you for the opportunity to review the manuscript entitled, "Upper Cervical Manipulation and Manual Massage Do Not Modulate Sympatho-Vagal Balance or Blood Pressure in Women: A Randomized, Placebo-Controlled Clinical Trial." After re-reading the manuscript with the corrections made by the authors and after reviewing the responses to the reviews, I believe that the article in this form is suitable for publication in a journal.

We sincerely thank and appreciate all the effort dedicated to making this article more robust and coherent.